# Gut microbiota-derived tryptamine and phenethylamine impair insulin sensitivity in metabolic syndrome and irritable bowel syndrome

Lixiang Zhai[1,2,10], Haitao Xiao[3,10], Chengyuan Lin[1,10], Hoi Leong Xavier Wong [2,10], Yan Y. Lam [1,10], Mengxue Gong[4], Guojun Wu [5], Ziwan Ning[1,2], Chunhua Huang[1,2], Yijing Zhang[2], Chao Yang[6], Jingyuan Luo[1,2], Lu Zhang[6], Ling Zhao[7], Chenhong Zhang [4], Johnson Yiu-Nam Lau[2], Aiping Lu [2], Lok-Ting Lau[2], Wei Jia [8,9] ✉, Liping Zhao [5] ✉ & Zhao-Xiang Bian [1,2] ✉

The incidence of metabolic syndrome is significantly higher in patients with irritable bowel syndrome (IBS), but the mechanisms involved remain unclear. Gut microbiota is causatively linked with the development of both metabolic dysfunctions and gastrointestinal disorders, thus gut dysbiosis in IBS may contribute to the development of metabolic syndrome. Here, we show that human gut bacterium *Ruminococcus gnavus*-derived tryptamine and phenethylamine play a pathogenic role in gut dysbiosis-induced insulin resistance in type 2 diabetes (T2D) and IBS. We show levels of *R. gnavus*, tryptamine, and phenethylamine are positively associated with insulin resistance in T2D patients and IBS patients. Monoassociation of *R. gnavus* impairs insulin sensitivity and glucose control in germ-free mice. Mechanistically, treatment of *R. gnavus*-derived metabolites tryptamine and phenethylamine directly impair insulin signaling in major metabolic tissues of healthy mice and monkeys and this effect is mediated by the trace amine-associated receptor 1 (TAAR1)-extracellular signal-regulated kinase (ERK) signaling axis. Our findings suggest a causal role for tryptamine/phenethylamine-producers in the development of insulin resistance, provide molecular mechanisms for the increased prevalence of metabolic syndrome in IBS, and highlight the TAAR1 signaling axis as a potential therapeutic target for the management of metabolic syndrome induced by gut dysbiosis.

Irritable bowel syndrome (IBS) is a common functional bowel disorder that is characterized by bowel habits changes and recurrent abdominal pain[1]. Recent studies show IBS is related to a higher prevalence of metabolic syndrome and T2D, suggesting IBS is a risk factor for metabolic disorders[2–4]. However, the cause and molecular mechanism of IBS leading to metabolic syndrome are still unknown. The gut microbiome has been extensively studied in the past decades for its correlation and causality with metabolic disorders including metabolic syndrome, insulin resistance, and diabetes as well as gastrointestinal disorders, such as IBS[5–7]. Therefore, gut dysbiosis in gastrointestinal

disorders may contribute to the development of metabolic disorders, but the pathogenic role and underlying molecular mechanisms of gastrointestinal disorders-related gut dysbiosis in the development of metabolic disorders are not elucidated yet.

Changes in gut microbiota composition in patients with metabolic disorders have been shown to induce changes in the content of gut-microbial products including lipopolysaccharides, short-chain fatty acids, bile acids, trimethylamine N-oxide and imidazole propionate, and these changes are widely recognized for their beneficial or detrimental effects on glucose tolerance in both animal and human studies[8,9]. In addition to these microbial metabolites, dietary amino acids including aromatic and branched-chain amino acids can also be catabolized by the gut microbiota into numerous metabolites with the potential effects on the metabolic health of hosts[10–12]. Our previous study has shown an anaerobic, gram-positive, and IBS-associated bacterium namely *Ruminococcus gnavus* produces tryptamine and phenethylamine by utilizing dietary amino acid tryptophan and phenylalanine to induce diarrheal symptoms in IBS patients[13]. Another recent study showed *R. gnavus* is positively associated with features of metabolic syndrome[14], but the causal effects of *R. gnavus* and its related pathogenetic factors involved in metabolic syndrome remain unknown.

In this study, we provided mechanistic insights into the contribution of gut microbes-derived tryptamine and phenethylamine to the development of insulin resistance in metabolic syndrome and IBS. Our findings in the present study identify *R. gnavus* and its-derived tryptamine and phenethylamine are positively associated with insulin resistance in patients with IBS. We demonstrated that monoassociation of *R. gnavus* led to insulin resistance and glucose intolerance concomitantly with elevated fecal tryptamine and phenethylamine levels in germ-free mice. Mechanistically, gut microbes-derived tryptamine and phenethylamine impair insulin signaling in the metabolic tissues via activation of the trace amine-associated receptor 1 (TAAR1)-extracellular signal-regulated kinase (ERK) signaling axis. Pharmacological antagonism of TAAR1 protects against insulin resistance in antibiotic-treated mice colonized with *R. gnavus* or transplanted with fecal microbiota from IBS patients. We further showed fecal tryptamine and phenethylamine levels were both positively correlated with glucose intolerance in patients with T2D and negatively correlated with the improvement of insulin sensitivity in T2D patients in a dietary fiber intervention study, suggesting tryptamine and phenethylamine are a comorbid factor of IBS and insulin resistance that caused by gut dysbiosis. Not only did we provide insights into the causal role of gut dysbiosis-associated insulin resistance in IBS, but also construct fundamental knowledge for developing gut microbiota-based therapeutics for the management of metabolic syndrome.

## Results

### The positive association between *R. gnavus*-derived tryptamine/phenethylamine and insulin resistance in irritable bowel syndrome

We first determined the levels of fasting blood glucose (FBG), triglyceride (TG), and triglyceride-glucose index (TyG), a marker of insulin resistance, in IBS patients and healthy controls (HC). Interestingly, we found fasting blood glucose, triglyceride, and triglyceride-glucose (TyG) index are all significantly increased in IBS patients ($p \le 0.001$ in all cases, Fig. 1A–C), suggesting IBS patients have higher diabetes risks compared with healthy controls. We then analyzed the association between gut microbes and insulin resistance in HC and IBS participants using their shotgun metagenomic sequencing data[15] and an insulin resistance marker TyG index (Fig. S1A). Among characterized gut microbes, *R. gnavus* was found positively correlated with TyG ($r = 0.2$, $p = 0.034$ for HC and $r = 0.186$, $p < 0.001$ for IBS, Fig. 1D, E) in HC and IBS patients and enriched in IBS patients with higher TyG index ($p = 0.03$, Fig. S1B), which was similar as a recent population-based

clinical study showing that *R. gnavus* is associated with several features of metabolic syndrome including increased levels of serum triglyceride and hemoglobin A1c (HbA1c)[14]. Our previous result has shown *R. gnavus* is a major tryptamine and phenethylamine producer[13], we also found fecal tryptamine and phenethylamine levels are positively correlated with TyG in HC and IBS patients ($r = 0.196$ and $0.255$, $p < 0.001$, Fig. S1C) and significantly enriched in IBS patients with higher TyG index ($p < 0.05$ in both, Fig. S1D, E).

### Colonization with *R. gnavus* impaired insulin sensitivity accompanied by tryptamine and phenethylamine overproduction

To evaluate the pathogenic role of *R. gnavus* in metabolic syndrome, we colonized a human gut bacterium strain *R. gnavus* (ATCC 29149) in germ-free mice. Despite no significant changes in body weight (Fig. S1F), germ-free mice colonized with *R. gnavus* exhibited impaired glucose tolerance and reduced insulin sensitivity as determined by oral glucose tolerance test (OGTT) and insulin tolerance test (ITT) ($p < 0.05$ in all cases, Fig. 1F, G).

Given that *R. gnavus* is a primary gut microbe that is capable of decarboxylating tryptophan and phenylalanine into tryptamine and phenethylamine as shown by our previous findings and other studies[13,16], elevated fecal tryptamine and phenethylamine levels were found along with impaired glucose tolerance and insulin sensitivity in germ-free mice colonized with *R. gnavus* ($p < 0.01$ in all cases, Fig. 1H, I). To further investigate whether tryptamine and phenethylamine produced by *R. gnavus*-derived decarboxylase (TDC) impair insulin sensitivity in vivo, we ectopically expressed TDC gene from *R. gnavus* (ATCC 29149) in a commensal gram-positive bacterium *Lactobacillus casei*, which does not produce tryptamine or phenethylamine. We showed that both tryptamine and phenethylamine levels were significantly increased in the culture medium from the engineered *L. casei* TDC⁺ strain when compared to the *L. casei* with empty vector and medium control after overnight inoculation (Fig. S1G, H). Colonization of *L. casei* TDC⁺ strain induced higher OGTT and ITT indexes together with elevated tryptamine and phenethylamine levels in antibiotics-treated mice compared with blank vector *L. casei* ($p < 0.05$ in all cases, Fig. 1J–M and Fig. S1I), indicating that tryptamine and phenethylamine produced from *R. gnavus*-derived TDC may impair insulin sensitivity in vivo.

### The positive association between tryptamine/phenethylamine and glucose intolerance in type 2 diabetes

We further determined the fecal levels of tryptamine, phenethylamine, and their precursors (tryptophan and phenylalanine) in healthy controls and subjects with T2D. We found that tryptamine and phenethylamine were significantly higher in fecal samples from T2D subjects ($p = 0.011$ and $0.031$, Fig. 2A, B). In contrast, the levels of tryptophan and phenylalanine were not altered in fecal samples of T2D subjects (Fig. S2A, B). In addition, the levels of tyrosine and tyramine, as one of the aromatic trace amines (tryptamine, phenethylamine, and tyramine), were not found significantly altered in T2D subjects (Fig. S2C, D). Moreover, correlation analyses revealed fecal tryptamine and phenethylamine were positively correlated with fasting blood glucose (FBG) levels ($r = 0.443$ and $0.378$, $p = 0.005$ and $0.008$, Fig. 2C, D) in T2D patients, revealing tryptamine and phenethylamine exhibit a positive association with glucose intolerance.

Host variables, such as age, anti-diabetic medications, and dietary patterns, may confound gut microbiota studies of human diseases[17]. To reduce the influence of these variables, we also determined fecal tryptamine, phenethylamine, and tyramine levels in monkeys (*Macaca fasicularis*) with spontaneous metabolic syndrome, a pre-clinical primate model for metabolic diseases[18]. Based on FBG and HbA1c levels[19], age-matched and treatment-naive monkeys were assigned to normal group, pre-diabetes or diabetes groups (Fig. S2E–I). We showed that

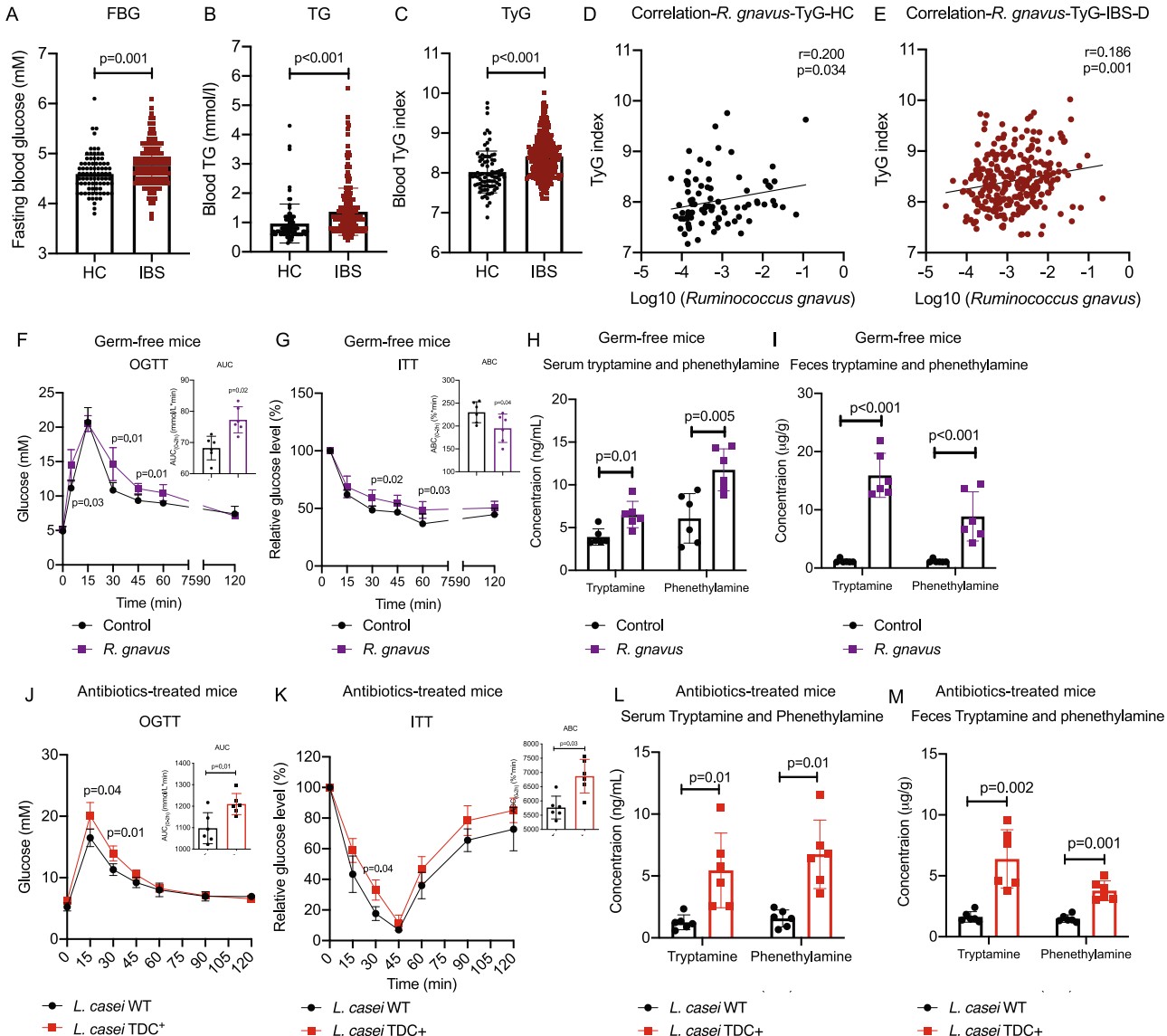

**Fig. 1 | Positive association and causality between *R. gnavus*-derived tryptamine/phenethylamine and insulin resistance in irritable bowel syndrome.** **A**–**C** FBG, TG, and TyG levels in IBS-D patients (*n* = 290) compared with healthy control (HC, *n* = 89) subjects (determined by two-tailed *t*-tests). **D**, **E** Spearman's correlation between relative abundances of *R. gnavus* and TyG levels in HC and IBS-D subjects (determined by one-tailed test). **F**, **G** OGTT and ITT indexes in germ-free mice following colonization of *R. gnavus* ATCC 29149 (*n* = 6 per group) (determined by two-tailed *t*-tests). **H**, **I** Tryptamine and phenethylamine levels in serum and fecal samples of germ-free mice following colonization of *R. gnavus* ATCC 29149 (*n* = 6

per group) (determined by two-tailed *t*-tests). **J**, **K** OGTT and ITT indexes in antibiotics-treated mice following colonization of either engineered *L. casei* TDC$^+$ or *L. casei* WT (*n* = 6 per group) (determined by two-tailed *t*-tests). **L**, **M** Tryptamine and phenethylamine levels in serum and fecal samples of antibiotics-treated mice following colonization of either engineered *L. casei* TDC$^+$ or *L. casei* WT for 3 days (*n* = 6 per group) (determined by two-tailed *t*-tests). *P* values were determined by ordinary two-way ANOVA or Student's *t*-test. Data are presented as mean ± S.D. See additional data in Fig. S1.

mice fed with fecal suspensions from diabetic monkeys exhibited higher glucose levels than mice with fecal suspensions from normal monkeys (*p* < 0.05 in all cases, Fig. S2J), suggesting a causal role of gut microbiota and their metabolic products from diabetic monkeys in the development of glucose intolerance. We then quantified tryptamine, phenethylamine, tyramine, and their precursors (tryptophan, phenylalanine, and tyrosine) in the sera and feces of the monkeys. Consistently, tryptamine and phenethylamine were significantly increased in sera and feces of diabetic monkeys (*p* < 0.05 in all cases, Fig. 2E, F), whereas fecal levels of tryptophan, phenylalanine, tyrosine, and tyramine were not significantly changed (Fig. S2K–N). Correlation analyses revealed that fecal/serum levels of tryptamine and phenethylamine were all positively correlated with HbA1c (*r* = 0.253 and 0.245, *p* < 0.05 in all cases) (Fig. 2G, H) and FBG (*r* = 0.275/0.255

and *r* = 0.452/0.368, *p* < 0.02 in all cases) (Fig. 2I, J and Fig. S2O, P) in monkeys. We also found higher concentrations of tryptamine and phenethylamine in the culture medium of gut bacteria from fecal samples of pre-diabetic and diabetic monkeys under anaerobic conditions (*p* = 0.006 and 0.02, Fig. 2K, L), confirming that diabetes-associated microbiota has a higher catalytic ability to transform tryptophan and phenylalanine into tryptamine and phenethylamine in monkeys.

Our previous studies have reported that dietary fiber intervention improved glucose homeostasis in T2D subjects by promoting the growth of short-chain fatty acids-producing bacteria[20,21]. Interestingly, we identified *R. gnavus* CAG0075 is significantly suppressed toward dietary fiber intervention in our clinical study (Supplementary Data 1). The TDC sequence of *R. gnavus* CAG0075 had 100% identity compared

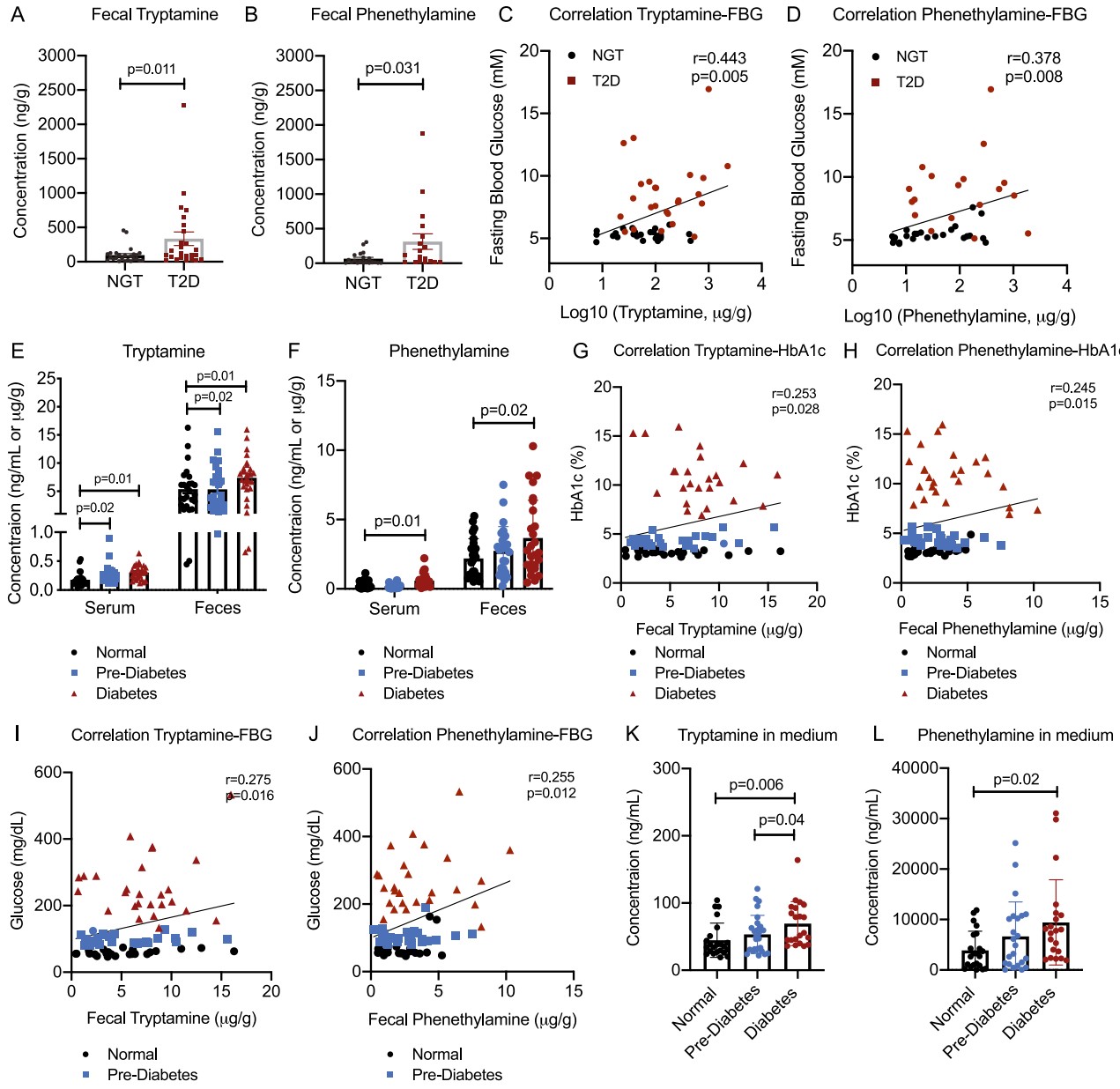

**Fig. 2 | Tryptamine and phenethylamine are positively correlated with glucose intolerance in patients with type 2 diabetes and monkeys with spontaneous diabetes. A, B** Tryptamine and phenethylamine levels in fecal samples of individuals with or without T2D ($n = 25$ subjects with NGT, $n = 25$ patients with T2D). **C, D** Spearman's correlation between fecal tryptamine/phenethylamine levels with FBG level in individuals with or without T2D ($n = 25$ subjects with NGT, $n = 25$ patients with T2D). **E, F** Tryptamine and phenethylamine levels in serum and fecal samples of age-matched monkeys with or without pre-diabetes and diabetes ($n = 26$/group). **G, H** Spearman's correlation between fecal tryptamine/

phenethylamine levels with HbA1c index in monkeys with or without pre-diabetes and diabetes ($n = 26$/group). **I, J** Spearman's correlation between fecal tryptamine/phenethylamine levels with FBG index in monkeys with or without pre-diabetes and diabetes ($n = 26$/group). **K, L** Tryptamine and phenethylamine production in batch culture experiments using feces from monkeys with or without pre-diabetes and diabetes ($n = 26$/group). $P$ values were determined by one-tailed ordinary two-way ANOVA or Student's $t$-test. Data are presented as mean ± S.D. See additional information in Figure. S2.

with the reference TDC sequence from *R. gnavus* (ATCC 29149)[16] and the average nucleotide identity between *R. gnavus* CAG0075 and *R. gnavus* (ATCC 29149) was 99.1%, highlighting the *R. gnavus* TDC is negatively correlated with the improvement of insulin sensitivity in T2D subjects. We then determined the tryptamine and phenethylamine levels in fecal samples of T2D subjects who consumed a high-fiber diet in this study. Consistent with the reduction of *R. gnavus*, fecal tryptamine and phenethylamine levels and the ratio of tryptamine/tryptophan and phenethylamine/phenylalanine were significantly suppressed by the dietary fiber intervention (as shown in W group)

($p = 0.001$, Fig. 3A, B and Fig. S3A, B). In the high dietary fiber-treated T2D subjects (W group), downregulation of fecal tryptamine and phenethylamine levels were found positively correlated with improvements of HbA1c and HOMA-IR indexes ($r = 0.569/0.269$ and $r = 0.544/0.356$ after adjustment to BMI, $p < 0.05$ in all cases, Fig. 3C, D and Fig. S3C, D). In addition, correlation analyses between tryptamine/phenethylamine and bacterial genomes with the TDC sequence showed that only *R. gnavus* CAG0075 was positively correlated with tryptamine among the 5 genomes with the TDC genes (Supplementary Data 2). These results suggest that the *R. gnavus* accompanied with

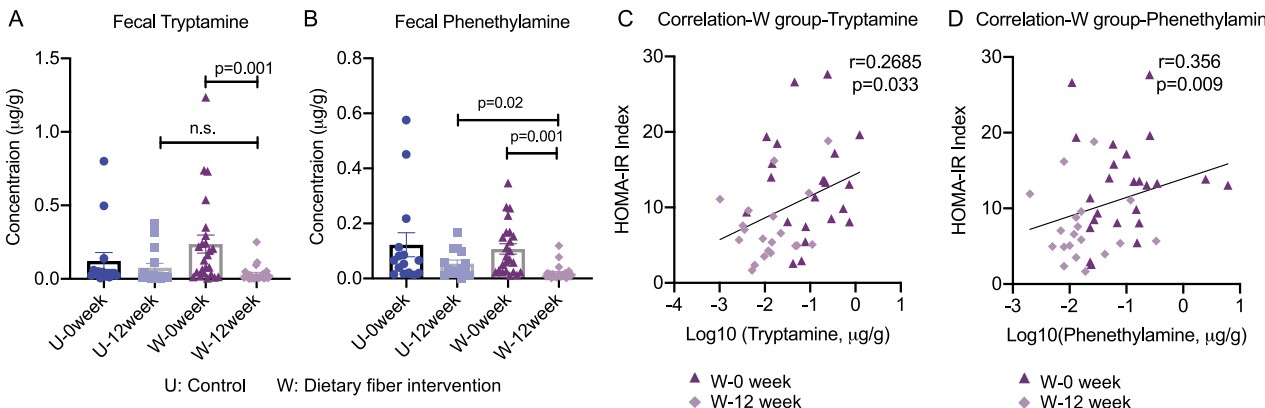

**Fig. 3 | Tryptamine and phenethylamine are negatively correlated with the improvement of insulin sensitivity in dietary fiber-treated patients with type 2 diabetes. A**, **B** Tryptamine and phenethylamine levels in fecal samples from T2D subjects in the control group (U group; n = 16) and high-fiber group (W group; n = 27) (Day 0 and Day 84). **C**, **D** Spearman's correlation analysis between fecal tryptamine and phenethylamine level and HOMA-IR index in T2D subjects consuming a high fiber diet (W group; n = 27). Differences in phenethylamine and tryptamine levels in serum and fecal samples were analyzed by one-tailed ordinary two-way ANOVA or Student's t-test. Data are presented as mean ± S.D. See additional information in Fig. S3.

tryptamine and phenethylamine are negatively correlated with improvement of insulin sensitivity in T2D subjects.

### Tryptamine and phenethylamine impair insulin sensitivity

To understand the role of tryptamine and phenethylamine in the pathophysiology of insulin resistance, we administered tryptamine and phenethylamine into normal mice within the pathophysiological range detected in the fecal samples of IBS subjects and T2D subjects. In normal mice intraperitoneally injected with tryptamine or phenethylamine, we found remarkably inhibitory effects of tryptamine and phenethylamine on glucose tolerance and insulin sensitivity observed by OGTT and insulin tolerance test (ITT) ($p < 0.05$ in all cases, Fig. 4A–D). Moreover, tryptamine and phenethylamine significantly suppressed the insulin-induced Akt phosphorylation in major metabolic tissues, including white adipose tissue (WAT), liver, and skeletal muscle ($p < 0.05$ in all cases, Fig. 4E–H). We also showed tryptamine and phenethylamine induced a significant elevation of serum TG levels in normal mice ($p < 0.05$ for both, Fig. S4A, B). To validate our findings in mice, we used an intravenous glucose tolerance test (IVGTT) to determine whether glucose tolerance was similarly impaired by tryptamine in normal monkeys at a single dose of 10 mg/kg via oral gavage. Consistently, we showed that tryptamine-treated monkeys exhibited higher blood glucose levels in the glucose challenge ($p < 0.05$, Fig. S4C). Moreover, we found a significant elevation of serum insulin levels in both tryptamine and phenethylamine-treated monkeys and mice ($p < 0.01$ in all cases, Fig. S4D and Fig. S4K).

In line with in vivo studies, we showed that treatment of tryptamine and phenethylamine inhibited insulin signaling in a time-dependent manner in 3T3-L1 adipocytes, a valid human cell line for the study of insulin actions ($p < 0.05$ in all cases, Fig. 4I–L). We also showed tryptamine inhibited basal glucose uptake in a dose-dependent manner in 3T3-L1 adipocytes ($p < 0.05$ in all cases, Fig. S4E). In contrast, treatment with precursors and metabolites of tryptamine and phenethylamine including tryptophan, phenylalanine, indole-3-acetic acid, and phenylacetic acid in similar doses did not alter the insulin-induced AKT phosphorylation in 3T3-L1 cells (Fig. S4F, G), suggesting that tryptamine and phenethylamine but not their precursors or metabolites impair insulin sensitivity. Moreover, we showed acute treatment of tryptamine and phenethylamine did not significantly affect the serum levels of GLP-1 and PYY (Fig. S4H, I).

### Tryptamine and phenethylamine weaken insulin signaling via TAAR1-ERK activation

To investigate the mechanism(s) underlying tryptamine and phenethylamine inhibition on insulin signaling, we employed a phospho-proteomics approach to capture the molecular components that are significantly altered by tryptamine and phenethylamine (Supplementary Data 6–7). After oral gavage of tryptamine, we observed that tryptamine and its metabolite indole-3-acetic acid (IAA) levels were significantly increased in serum and insulin-sensitive tissues within 15 min ($p < 0.05$ in all cases, Fig. S5A, B), suggesting that tryptamine enters insulin-sensitive tissues and is simultaneously metabolized by the host after being produced by gut microbiota. We then found four insulin signaling-related proteins, including hormone-sensitive lipase (HSL), mitogen-activated protein kinase MAPK 1/3 (ERK), sorbin, and SH3 domain containing 1 (SH3D5), were upregulated in insulin-sensitive tissues in response to tryptamine treatment (Supplementary Data 3). Among these proteins, ERK has previously been implicated in the pathogenesis of IR in T2D[22]. To test whether tryptamine and phenethylamine suppress insulin signaling through activation of the MAPK/ERK pathway, we examined ERK1/2 phosphorylation in tryptamine-treated mice (i.p.) and found that ERK1/2 phosphorylation is upregulated in insulin-sensitive tissues by tryptamine ($p < 0.05$ in all cases, Fig. 5A, B). Next, we used ERK inhibitors U0126 and PD98059 to determine whether the suppressive effect of tryptamine on glucose tolerance and insulin sensitivity is dependent on the MAPK/ERK pathway. In OGTT and ITT studies, treatment with ERK inhibitors significantly improved glucose intolerance and insulin resistance in tryptamine and phenethylamine-treated mice ($p < 0.05$ in all cases, Fig. 5C, D, Fig. S5C and Fig. 5G, H). By contrast, ERK inhibitors exerted negligible effects on glucose tolerance and insulin sensitivity in control mice. In addition, ERK inhibitors abolished the inhibitory effects of tryptamine on the insulin-stimulated AKT phosphorylation and significantly downregulated tryptamine-induced ERK phosphorylation in insulin-sensitive tissues ($p < 0.05$ in all cases, Fig. 5E, F and Fig. 5I, J). In line with these in vivo observations, tryptamine and phenethylamine also induced ERK1/2 phosphorylation in a time-dependent manner in 3T3-L1 cells, which was blocked by pre-treatment with ERK inhibitors ($p < 0.05$ in all cases, Fig. S5D–F). ERK inhibitors also significantly suppressed the inhibitory effects of tryptamine and phenethylamine on AKT phosphorylation in insulin-stimulated 3T3-L1 cells ($p < 0.05$ in all cases, Fig. S5G, H). These data suggested that tryptamine and phenethylamine impair insulin sensitivity by activating the MAPK/ERK pathway.

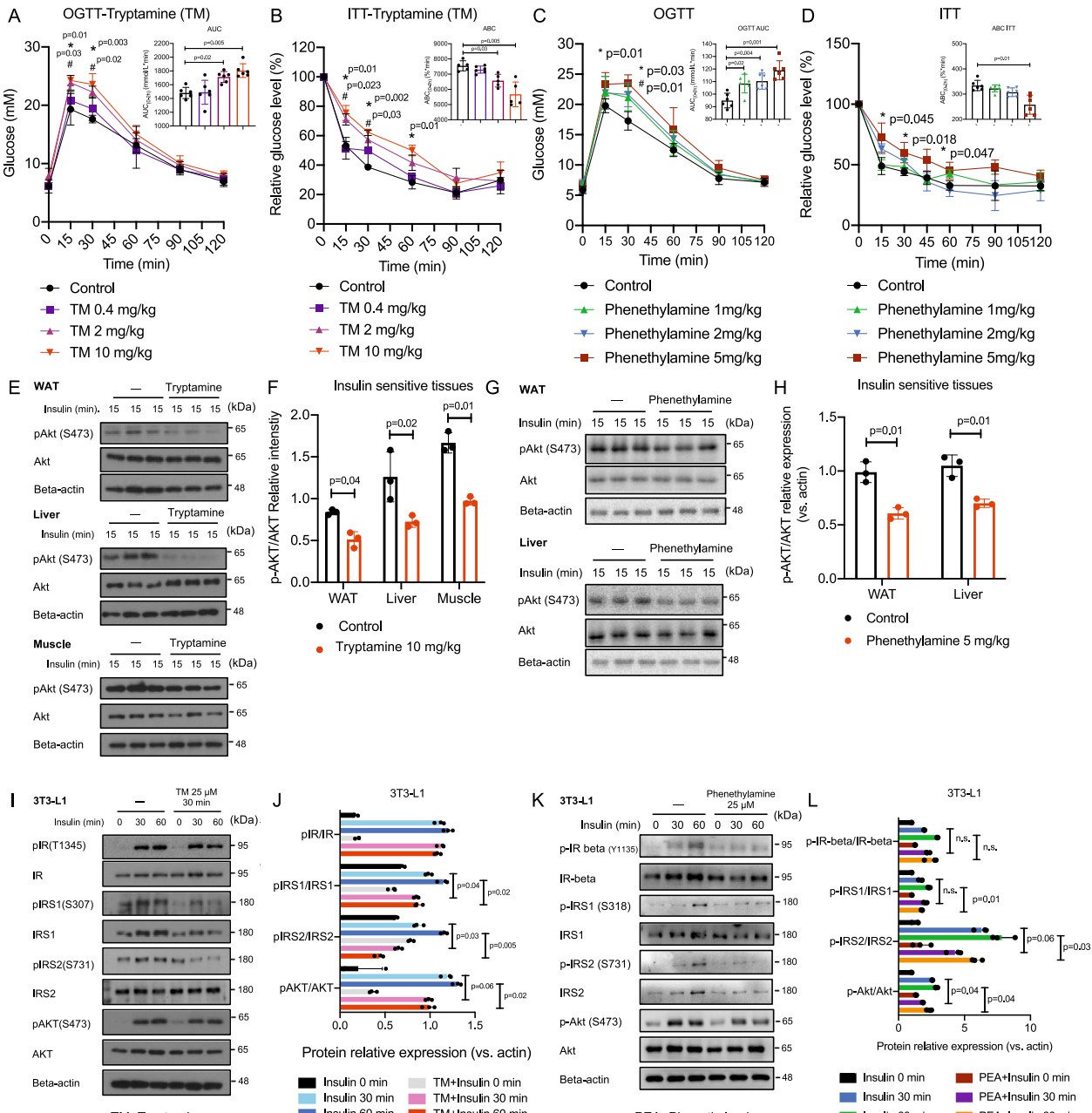

**Fig. 4 | Tryptamine and phenethylamine impair insulin sensitivity in mice, monkeys, and in vitro models. A, B** OGTT and ITT in normal mice after treatment with tryptamine at indicated dosages (0.4 mg/kg, 2 mg/kg, and 10 mg/kg) or control (1% DMSO in saline) by i.p. ($n = 6$/group). *comparison between the control group and tryptamine group (10 mg/kg). #comparison between the control group and tryptamine-treated group (2 mg/kg). **C, D** OGTT and ITT in normal mice after treatment with phenethylamine at indicated dosages (1 mg/kg, 2 mg/kg, and 5 mg/kg) or control (1% DMSO in saline) by i.p. ($n = 6$/group). *comparison between the control group and phenethylamine group (5 mg/kg). #comparison between the control group and tryptamine-treated group (2 mg/kg). **E, F** Western blot (and quantification) of the effect of tryptamine treatment (10 mg/kg) by i.p. on AKT phosphorylation stimulated by insulin (1 U/kg) in WAT lysates, liver lysates and skeletal muscle lysates from mice. ($n = 3$/group). **G, H** Western blot (and quantification) of the effect of phenethylamine treatment (5 mg/kg) by i.p. on AKT phosphorylation stimulated by insulin (1 U/kg) in WAT lysates and liver lysates from mice. ($n = 3$/group). **I, J** Western blot (and quantification) of the effect of tryptamine treatment (25 μM) on insulin signaling stimulated by insulin (20 nM) in 3T3-L1 cells ($n = 3$/group). **K, L** Western blot (and quantification) of the effect of phenethylamine treatment (25 μM) on insulin signaling stimulated by insulin (20 nM) in 3T3-L1 cells ($n = 3$/group). Data are presented as mean ± S.D. $P$ values were determined by two-tailed ordinary one-way ANOVA or Student's $t$-test. See additional information in Fig. S4.

Both tryptamine and phenethylamine can bind to and activate GPCR receptor TAAR1[23,24], which may exert inhibitory effects on the downstream insulin signaling pathway. Furthermore, we sought to determine whether tryptamine and phenethylamine act through the TAAR1-MAPK/ERK signaling axis to inhibit insulin signaling. In the OGTT and ITT studies, treatment with EPPTB, a specific TAAR1 antagonist, significantly reduced tryptamine and phenethylamine-induced glucose intolerance and insulin resistance in mice ($p < 0.05$ in all cases, Fig. 6A–D). TAAR1 antagonism by EPPTB also downregulated tryptamine-induced ERK phosphorylation and abolished the inhibitory effects of tryptamine on insulin-stimulated AKT phosphorylation in insulin-sensitive tissues of mice ($p < 0.05$ in all cases, Fig. 6E, F and Fig. S6A, B). In contrast, EPPTB affects neither glucose tolerance nor insulin

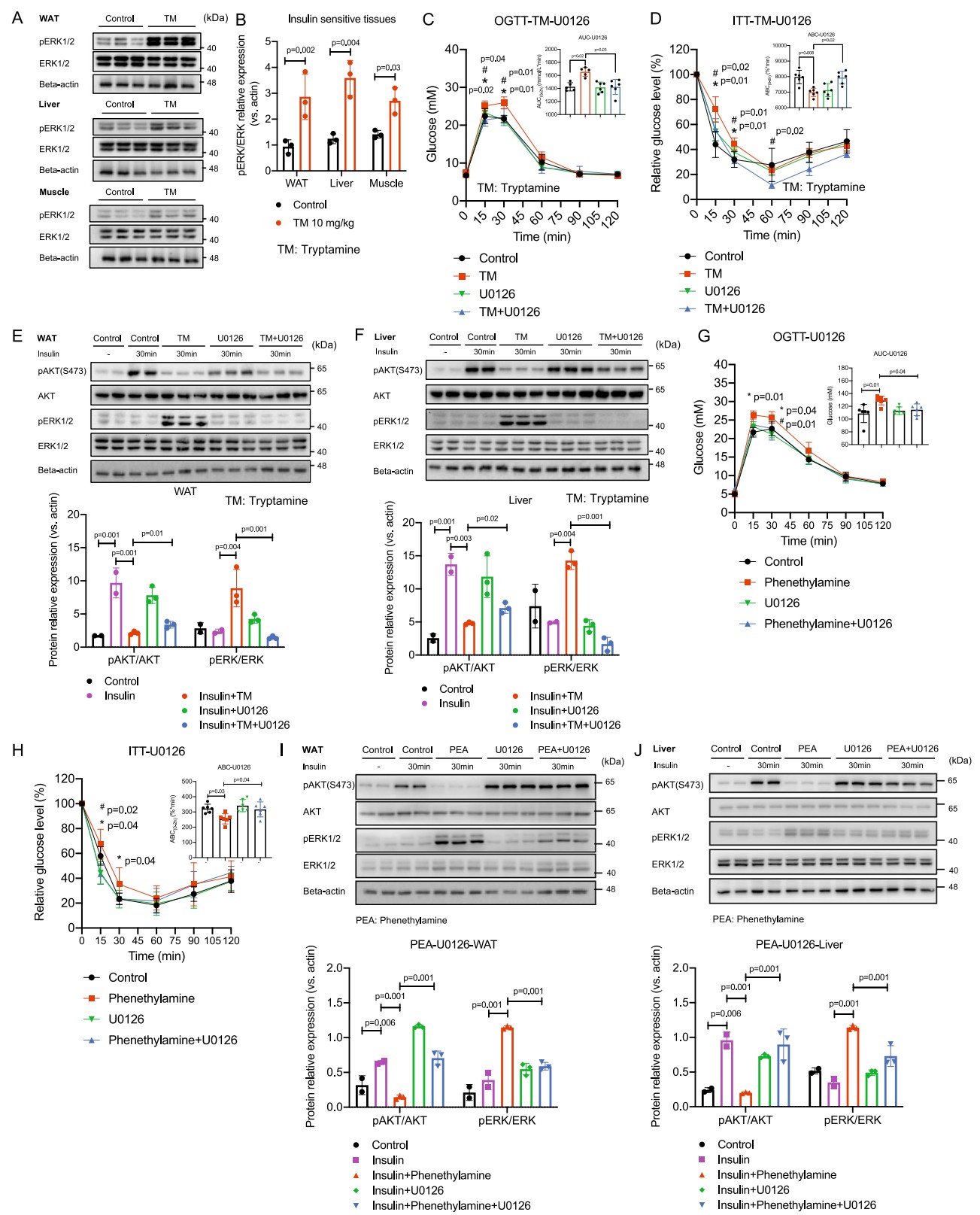

sensitivity in control mice. In line with these in vivo results, EPPTB treatment also significantly downregulated the increased ERK phosphorylation induced by tryptamine and phenethylamine ($p < 0.05$ in all cases, Fig. S6C and E) and reversed the inhibitory effects of tryptamine and phenethylamine on insulin-stimulated AKT phosphorylation in 3T3-L1 cells ($p < 0.05$ in all cases, Fig. S6D and F). Notably, genetic ablation of *Taar1* significantly suppressed

the expression of *Taar1* in metabolic tissues ($p < 0.001$ in all cases, Fig. 6G) and protected against tryptamine and phenethylamine-induced glucose intolerance and insulin resistance in mice ($p < 0.05$ in all cases, Fig. 6H–K).

**TAAR1 inhibition alleviate insulin resistance induced by gut dysbiosis.** We then investigated the pharmacological antagonism of

**Fig. 5 | Tryptamine and phenethylamine impair insulin signaling via ERK activation. A**, **B** Western blot (and quantification) of the effect of tryptamine treatment (10 mg/kg) on ERK activation in WAT lysates, liver lysates, and skeletal muscle lysates from mice ($n = 3$/group). **C**, **D** OGTT and ITT indexes in mice after treatment with tryptamine (10 mg/kg), ERK inhibitor U0126 (20 mg/kg), or control (1% DMSO in saline) ($n = 6$/group) by i.p. *comparison between the control group and tryptamine group (10 mg/kg). #comparison between tryptamine group and tryptamine + ERK inhibitor (U0126) group. **E**, **F** Western blot (and quantification) of the effects of tryptamine (10 mg/kg) and ERK inhibitor U0126 treatment (20 mg/kg) on ERK activation and insulin (1 U/kg)-stimulated AKT activation in WAT lysates and liver lysates from mice ($n = 2$ in control and insulin group, $n = 3$ in other groups) by i.p. **G**, **H** OGTT and ITT indexes in mice after treatment with phenethylamine (5 mg/kg), ERK inhibitor U0126 (20 mg/kg), or control (1% DMSO in saline) ($n = 6$/group) by i.p. *comparison between the control group and phenethylamine group (10 mg/kg). #comparison between tryptamine group and phenethylamine + ERK inhibitor (U0126) group. **I**, **J** Western blot (and quantification) of the effects of phenethylamine (5 mg/kg) and ERK inhibitor U0126 treatment (20 mg/kg) on ERK activation and insulin (1 U/kg)-stimulated AKT activation in WAT lysates and liver lysates from mice ($n = 2$ in control and insulin group, $n = 3$ in other groups) by i.p. Data are presented as mean ± S.D. $P$ values were determined by two-tailed ordinary one-way ANOVA or Student's $t$-test. See additional information in Fig. S5.

TAAR1 on glucose tolerance and insulin resistance in antibiotics-treated mice induced by gut dysbiosis including *R. gnavus* and fecal microbiota from IBS patients with high tryptamine and phenethylamine levels as well as high TyG index. In line with previous findings in germ-free mice, antibiotics-treated mice colonized with *R. gnavus* exhibited lower insulin sensitivity and impaired glucose tolerance, and these insulin-desensitizing effects of *R. gnavus* were partially abrogated by EPPTB treatment ($p < 0.05$ in all cases, Fig. 7A–C), indicating inhibition of TAAR1 alleviated *R. gnavus*-induced insulin resistance. We then conducted a fecal microbiota transplant study and showed fecal microbiota from IBS patients with high tryptamine and phenethylamine levels also exhibited glucose intolerance in antibiotics-treated mice ($p < 0.05$ in all cases, Fig. 7D and Fig. S7A). Following this finding, we showed genetic ablation of *Taar1* in mice also significantly inhibited the glucose intolerance induced by fecal microbiota from IBS patients with high TyG index and high tryptamine and phenethylamine levels ($p < 0.05$ in all cases, Fig. 7E and Fig. S7B).

## Discussion

Growing evidence suggests a causal link between the gut microbiome and human metabolic health. A recent cross-sectional study ($n = 5215$ in total) showed *R. gnavus* has the strongest association with features of metabolic syndrome among 50 identified prevalent gut microbes in the species level[14]. However, the causal effects of *R. gnavus* on metabolic syndrome and its pathogenetic mechanisms have not been explored yet. Our findings demonstrating tryptamine/phenethylamine-mediated TAAR1 signaling pathway as the key molecular axis underlying *R. gnavus*-induced insulin resistance is of utmost importance and explained the molecular mechanisms of increased prevalence of metabolic syndrome in IBS.

In the present study, not only human subjects but also a preclinical monkey model of metabolic syndrome were involved to address the correlation between tryptamine/phenethylamine and insulin resistance for several reasons. First, unlike diet-, chemical- or genetically-induced animal models of T2D, the monkeys spontaneously develop a metabolic syndrome that highly resembles the major features of human metabolic syndrome characterized by hyperglycemia, hyperlipidemia, and insulin resistance. Second, the studies involving experimental monkeys are not influenced by confounding variables that can affect the gut microbiome in human studies, such as age, anti-diabetic medications, and dietary patterns. Gut microbiome composition may vary considerably across geographic locations, races, and ethnicities, while gut-microbial metabolites profiles are highly conserved[25], suggesting the use of the combined application of gut microbes and gut-microbial metabolites bring technical advantages to the prognosis and diagnosis of human diseases. Through administration of fecal suspension from normal and diabetic monkeys in HFD-fed mice, we showed gut dysbiosis in spontaneous diabetic monkeys also contributes to the development of glucose intolerance.

From a mechanism of action standpoint, we revealed tryptamine and phenethylamine derived from the *R. gnavus*-mediated catabolism on dietary amino acids impaired insulin sensitivity via activation of TAAR1-MAPK/ERK signaling pathway axis, thereby contributing to insulin resistance in gut dysbiosis-associated IBS and T2D. TAAR1 is an amine-activated G protein-coupled receptor that is activated by gut microbes-derived aromatic trace amines including tryptamine, phenethylamine, and tyramine in gut[26]. An endogenous activator (3-iodothyronamine) of TAAR1 has recently been shown to improve glycemic control by promoting insulin secretion in beta-cells[27]. In our present study, we also showed tryptamine and phenethylamine can stimulate insulin secretion but both acute and long-term exposure to TAAR1 activators tryptamine and phenethylamine impair insulin sensitivity, thereafter the pharmacological use of TAAR1 modulator on glucose control should be carefully considered. MAPK/ERK signaling pathway is involved in the development of insulin resistance associated with metabolic syndrome and T2D. ERK activity is elevated in WAT of humans and rodents in diabetic conditions[28,29] and activation of the ERK signaling pathway can significantly reduce the expression of key mediators of insulin signaling. In addition, inhibition of the ERK pathway using specific chemical inhibitors is effective in alleviating insulin resistance in *db/db* and HFD-fed mice[30]. Activation of TAAR1 leads to the activation of MAPK cascade including calmodulin-dependent protein kinase II and MAPK/ERK kinase 1/2 (MEK1/2)[27]. We demonstrated tryptamine and phenethylamine actions on ERK activation and inhibition of insulin signaling pathway can be abolished by TAAR1 antagonist and genetic ablation of *Taar1*, representing targeting TAAR1-MAPK/ERK signaling axis as a potential therapeutic strategy to combat *R. gnavus*-induced insulin resistance.

Besides TAAR1, several receptors of tryptamine including AhR and 5-HTR$_4$ have also been identified[31,32]. However, tryptamine-mediated activation of ERK was not suppressed by antagonists to either AhR or 5-HTR$_4$ in 3T3-L1 cells, suggesting TAAR1 plays a predominant role (Fig. S6G). Although tryptamine and its metabolite IAA are agonists of AhR signaling[32], the beneficial effects of AhR activation on the control of IR are likely predominantly mediated by IAA as the IAA concentration is about 40-100-fold to that of tryptamine in serum (Fig. S5A, B). Importantly, our findings revealed that IAA treatment did not impair glucose tolerance nor insulin sensitivity in normal mice (Fig. S4J), suggesting that the metabolic dysfunctions induced by alterations in the composition of the gut microbiome are primarily mediated by the tryptamine/TAAR1 signaling axis but not the IAA/AhR signaling axis in the context of T2D. We also determined the role of serotonin in the tryptamine and phenethylamine-induced insulin resistance and revealed the blockade of serotonin synthesis using a chemical inhibitor of TPH1 LX-1031 did not affect the inhibitory effects of acute treatment of tryptamine on glucose tolerance (Fig. S6H). In parallel with this in vivo finding, we showed tryptamine/phenethylamine directly impaired insulin signaling in a time-dependent manner in 3T3-L1 adipocytes, an in vitro cell line model that does not produce serotonin. However, as inhibition of peripheral serotonin biosynthesis reduces obesity and metabolic dysfunction[33] and the release of serotonin from intestinal EC cells into the circulation increases obesity and type 2 diabetes[34], serotonin also plays a vital role in regulating insulin sensitivity. Therefore, the stimulation of serotonin by tryptamine/phenethylamine may also contribute to the development of insulin

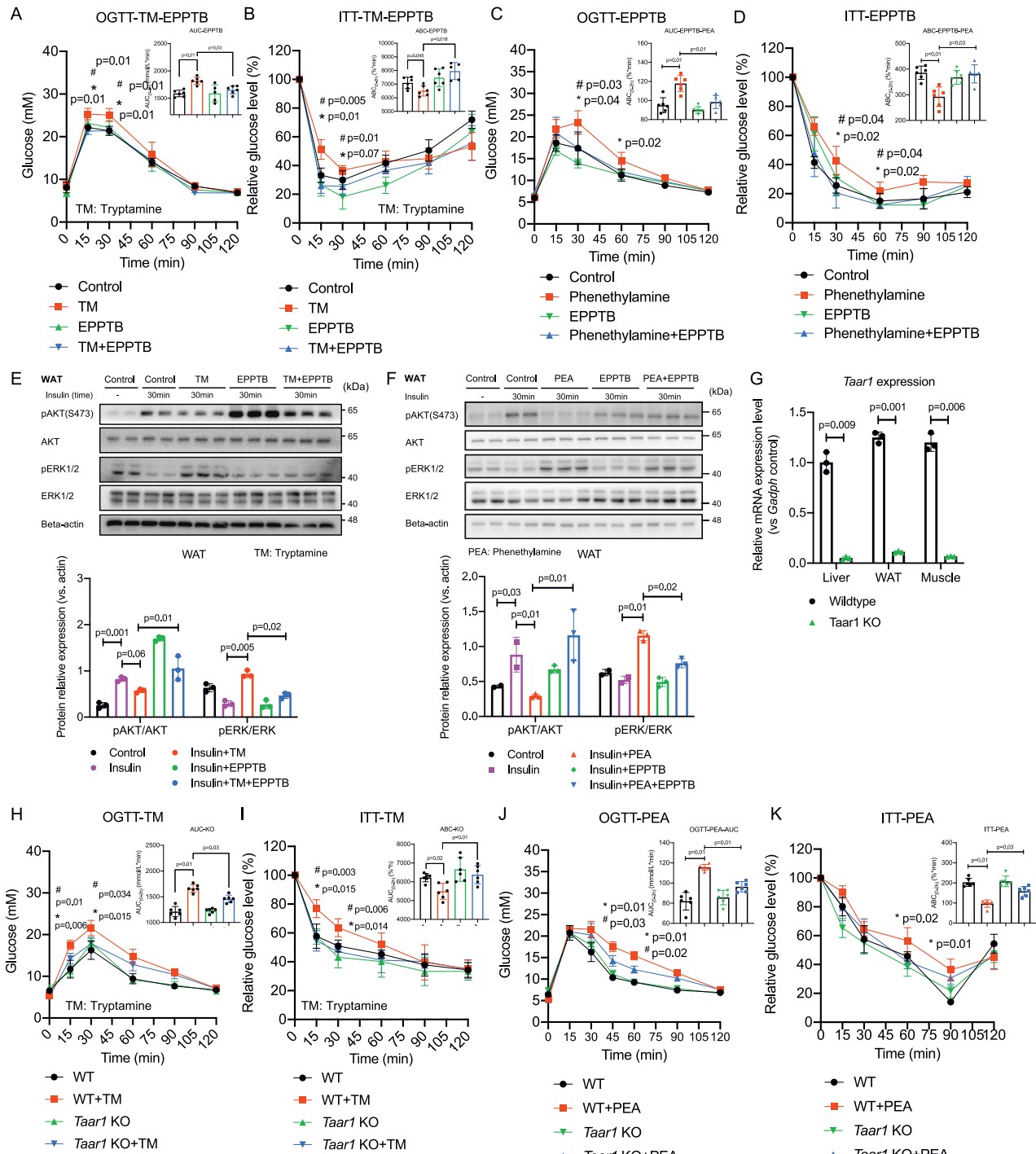

**Fig. 6 | Tryptamine and phenethylamine weaken insulin signaling via TAAR1-ERK signaling axis. A**, **B** OGTT and ITT indexes in mice after treatment with tryptamine (10 mg/kg), TAAR1 antagonist EPPTB (10 mg/kg), or control (1% DMSO in saline) by i.p. (*n* = 6/group). *comparison between the control group and tryptamine group (10 mg/kg). #comparison between tryptamine group and tryptamine +TAAR1 antagonist (EPPTB) group. **C**, **D** OGTT and ITT indexes in mice after treatment with phenethylamine (5 mg/kg), TAAR1 antagonist EPPTB (10 mg/kg), or control (1% DMSO in saline) by i.p. (*n* = 6/group). *comparison between the control group and phenethylamine group (5 mg/kg). #comparison between tryptamine group and phenethylamine+TAAR1 antagonist (EPPTB) group. **E**, **F** Western blot (and quantification) of the effects of tryptamine (10 mg/kg), phenethylamine (5 mg/kg), and TAAR1 antagonist EPPTB (10 mg/kg) treatment on ERK activation and insulin (1 U/kg)-stimulated AKT activation in WAT lysates from mice (*n* = 2 in

control and insulin group, *n* = 3 in other groups). **G** mRNA expression level of *Taar1* in insulin sensitive tissues of wildtype mice and *Taar1* KO mice (*n* = 3/group). **H**, **I** OGTT and ITT indexes in wildtype (WT) and *Taar1* knock out (KO) mice after treatment with tryptamine (10 mg/kg) or control (1% DMSO in saline) by i.p. (*n* = 6/ group). *comparison between with and without tryptamine treatment (10 mg/kg) in WT mice. #comparison between tryptamine treatment in WT and *Taar1* KO mice. **J**, **K** OGTT and ITT indexes in wildtype (WT) and *Taar1* knock out (KO) mice after treatment with phenethylamine (5 mg/kg) or control (1% DMSO in saline) by i.p. (*n* = 6/group). * comparison between with and without phenethylamine treatment (5 mg/kg) in WT mice. #comparison between tryptamine treatment in WT and *Taar1* KO mice. Data are presented as mean ± S.D. *P* values were determined by two-tailed ordinary one-way ANOVA or Student's t-test. See additional information in Figure S6.

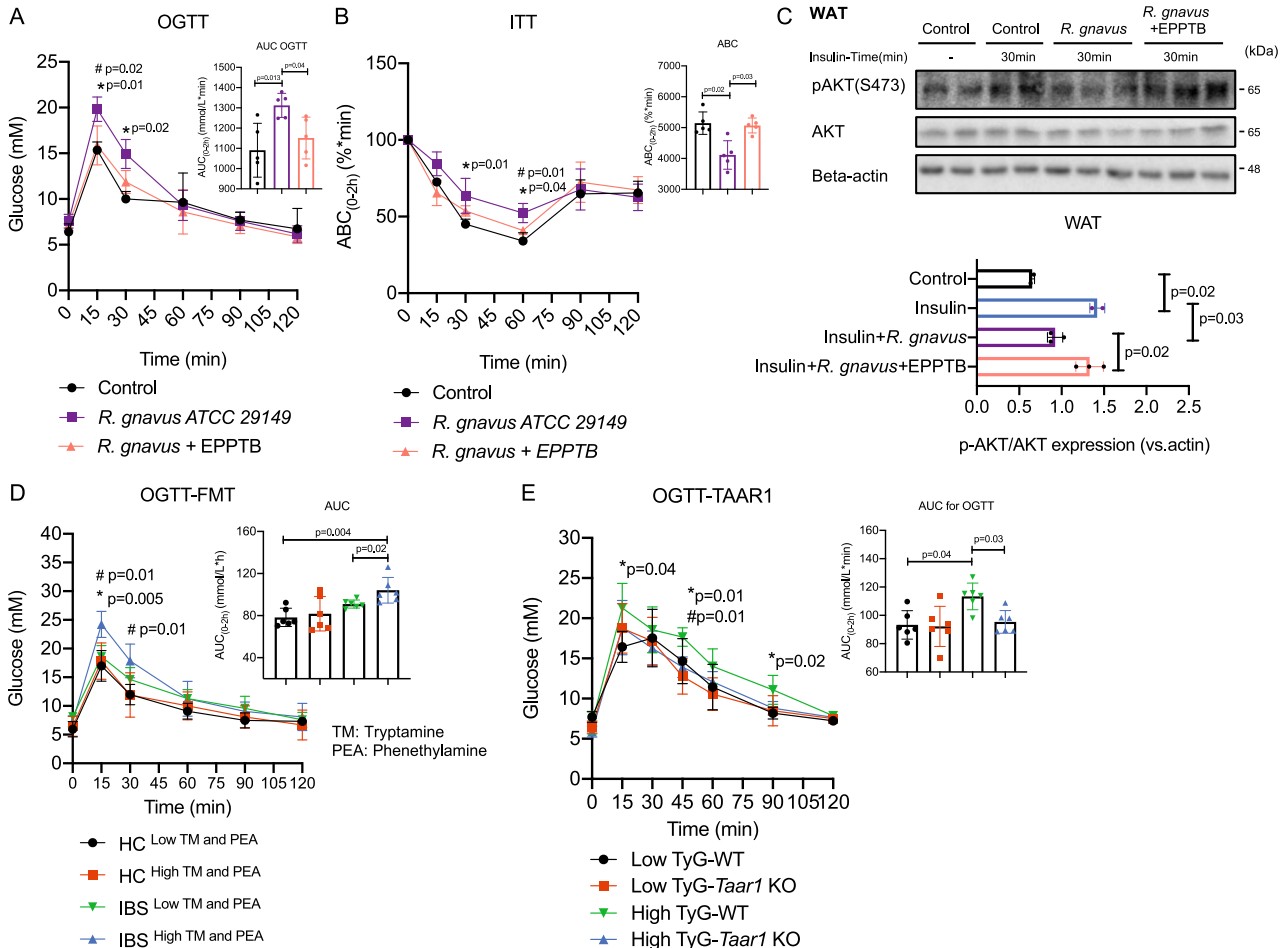

**Fig. 7 | TAAR1 inhibition alleviates insulin resistance induced by tryptamine and phenethylamine-producing bacteria. A, B** OGTT and ITT indexes in antibiotics-treated mice following colonization by *R. gnavus* ATCC 29149 and treatment with TAAR1 antagonist EPPTB (10 mg/kg) by i.p. (*n* = 6/group). *comparison between control group and tryptamine group (10 mg/kg). #comparison between tryptamine group and tryptamine+TAAR1 antagonist (EPPTB) group. **C** Western blot (and quantification) of effects of *R. gnavus* ATCC 29149 and TAAR1 antagonist EPPTB (10 mg/kg) on AKT activation stimulated by insulin (1 U/kg) in liver lysates from antibiotics-treated mice (*n* = 3 per group). **D** OGTT index in antibiotics-treated mice after transplantation with fecal microbiota from HC and IBS subjects with either low or high tryptamine and phenethylamine levels (*n* = 6/group). *comparison between HC low TM and PEA group and IBS high TM and PEA group. #comparison between IBS low TM and PEA group and IBS high TM and PEA group. **E** OGTT index in antibiotics-treated mice after transplantation with fecal microbiota from IBS subjects with either low or high TyG index (*n* = 6/group). Data are presented as mean ± S.D. *P* values were determined by two-tailed ordinary one-way ANOVA or Student's *t*-test. See additional information in Figure S7.

resistance in the long-term. This suggests that tryptamine and phenethylamine as TAAR1 ligands are the culprits and one possible way to design a therapeutic approach is to facilitate the reduction of *R. gnavus* or on top of blocking the tryptamine and phenethylamine production by inhibiting the bacterial TDC.

Besides the mechanistic insights into the contribution of *R. gnavus*-derived tryptamine and phenethylamine to the development of insulin resistance in IBS, a variety of bacteria species including *Blautia hansenii, Enterocloster* (*Clostridium*) *boltae* and *Enterococcus faecalis* and have also been shown to produce tryptamine and phenethylamine by recent studies[16,33]. Among these tryptamine and phenethylamine-producing bacteria species, *R. gnavus* has the highest catalytic ability to transform aromatic amino acids into aromatic trace amines (tryptamine, tyramine, and phenethylamine) compared with other bacteria species[33]. To further uncover the role of gut dysbiosis that is involved in the development of insulin resistance in IBS, we also investigated the associations between these bacteria, tryptamine/phenethylamine, and indicators of insulin resistance in IBS patients (Fig. S7C−H), providing supporting evidence for the increased abundances of other tryptamine and phenethylamine producer *B. hansenii* in metabolic syndrome. Given that bacterial TDC exists in many bacterial strains, the

therapeutic approach to regulate tryptamine and phenethylamine levels either by reducing the abundance of tryptamine and phenethylamine producers such as *R. gnavus* and *B. hansenii* or inhibiting tryptophan decarboxylase or blockade of tryptamine/TAAR1 signaling might be feasible.

Interestingly, our study showed dietary fiber intervention significantly suppressed tryptamine and phenethylamine levels and abundances of *R. gnavus* in T2D subjects, revealing manipulation of gut microbiota-derived tryptamine and phenethylamine by dietary changes or prebiotics is a potential direction for managing metabolic syndrome and IBS. We showed *R. gnavus* reduction accompanied by downregulation of tryptamine and phenethylamine are positively correlated with the improvement of insulin resistance. Given the insights on the interactions between foods, gut microbiota, and metabolic homeostasis, managing diet from a systemic perspective to reduce the risks of developing metabolic diseases may become an important therapeutic strategy. Further details about the mechanism are worthy to identify the proper management protocol from diet aspects such as a high-fiber diet or a low-tryptophan/phenylalanine diet to modulate the tryptamine and phenethylamine levels.

In summary, the data showed that *R. gnavus* and its derived tryptamine and phenethylamine are important factors in the pathogenesis of insulin resistance in both IBS and metabolic syndrome. Our further studies are going to determine the relative importance of these factors for their treatment and prediction potential for insulin resistance in different cohorts of metabolic disorders and gastrointestinal disorders, as well as in different geographic regions.

## Methods

### Monkey study

The first crab-eating macaques (*Macaca fascicularis*) metabolomics study is performed with ethics approval (No. YMB1704) by Yunnan Yinmore Biotechnology company (Kunming, China). The diagnosis of diabetes for monkeys was based on published criteria[34–36]. Specifically, age-matched (10-23 years old) monkeys were categorized as normal (Define as FBG < 75 mg/dL and Hb1Ac < 3.5%), pre-diabetes (FBG 80–130 mg/dL and Hb1Ac 4.0%–6.0%) and diabetes (FBG > 130 mg/dL and Hb1Ac > 6.0 %) (*n* = 26 for each group). Biological samples including serum and feces were collected and stored at −80 °C until analysis. The second crab-eating macaques (*n* = 5, 11–21 years old) intervention study is performed with ethics approval (No. HZ2021047) by Huazhen Biosciences company (Guangzhou, China). Biochemical parameters including OGTT and serum insulin level were measured before and after tryptamine treatment. No crab-eating macaques were treated with anti-diabetic medication during these studies.

### Mouse study

All mouse studies were approved by the Committee on the Use of Human & Animal Subjects in Teaching & Research at Hong Kong Baptist University (Hong Kong SAR, China) and performed following the Animals (Control of Experiments) Ordinance of the Department of Health, Hong Kong SAR, China and reported following the ARRIVE guidelines[37]. Male C57BL/6 J mice aged 6–8 weeks and weighing 20–25 g were purchased from the Laboratory Animal Services Centre, The Chinese University of Hong Kong (Hong Kong SAR, China). The mice were housed with a 12-h light/dark cycle at a controlled temperature of around 25 °C and a controlled humidity of 60% with free access to food and water.

Antibiotics-treated mice were generated using antibiotics cocktails containing 50 mg/kg vancomycin, 100 mg/kg neomycin, 100 mg/kg metronidazole, 100 mg/kg ampicillin, 50 mg/kg streptomycin via oral gavage for 9 days (one time per day) as previously reported[38]. Antibiotic cocktails administration were stopped 18 h before the fecal microbiota transplantation. During the fecal microbiota transplant experiment, the antibiotics were discontinued. Germ-free mice were purchased from Nanjing GemPharmatech Co. and the monoassociation study in germ-free mice was performed in sterile plastic isolators as previously reported[39].

*Taar1* (NM_053205) knockout mice (C57BL/6 J) were generated using CRISPR/Cas-mediated genome engineering and provided by Cyagen Biosciences (Suzhou) Inc. The *Taar1* knockout mice and their wildtype littermates were genotyped using PCR and southern blot by following primers (F1: GACAAAACGTAGTTGGAAGACTGA, R1: GTGTGCCTAGAAACCTTAACATCTG, R2: AATGTTTGTGATAGCGTGGCAAAG).

### Human study

The first cohort[40] study of healthy volunteers and IBS patients was approved by the Ethics Committee of Hong Kong Baptist University (HASC/15-16/0300 and HASC/16-17/0027). Written informed consent was signed and obtained from all participants. Subjects with morbid obesity or diabetes or with fasting blood glucose >7.0 mmol/L were excluded from this study.

The second cohort[41] of healthy controls and T2D subjects was approved by the Research Ethics Committee of Shanghai Jiao Tong University Affiliated Sixth People's Hospital. Written informed consent was signed and obtained from all participants. Subjects with fasting blood glucose <6.1 mmol/L were classified as healthy controls (HC), whereas those with fasting blood glucose >7.0 mmol/L or OGTT (2 h) >11.1 mmol/L were classified as T2D.

The third cohort GUT2D study[20] was approved by the Ethics Committee at the School of Life Sciences and Biotechnology, Shanghai Jiao Tong University (Ref ID: 2014-016). Written informed consent was obtained from all participants. The trial was registered in the Chinese Clinical Trial Registry (ChiCTR-TRC-14004959). Participants with T2D received either acarbose plus the usual diet (control; U group) or acarbose plus the WTP diet (intervention; W group) for 84 days.

All patients/public are not involved in the design, conduct, reporting, or dissemination plans of this study.

### Cell study

3T3-L1 adipocytes (ATCC CL-173) were cultured and maintained in DMEM with 10% (v/v) FBS. For the glucose uptake assay, 3T3-L1 cells were serum- and glucose-starved for 3 h and then incubated with glucose, FBS, insulin, and tryptamine for 30 min as indicated. To assess the effect of tryptamine on insulin signaling, 3T3-L1 cells were pretreated with or without tryptamine, ERK inhibitor, or TAAR1 antagonist EPPTB, and then treated with insulin at the indicated concentration. The effect of IAA on insulin signaling was evaluated by treating 3T3-L1 cells with or without IAA followed by insulin for the indicated times. Tryptamine, U0126, and EPPTB were dissolved in DMSO at 100 mM as a stock solution. For cAMP measurements, 3T3-L1 cells were pre-treated with or without EPPTB for 60 min and then treated with tryptamine for the indicated times.

### Bacterial strains culture

Tryptamine-producing *Lactobacillus casei* was constructed employing the tryptophan decarboxylase (TDC) gene from *Ruminococcus gnavus*. The TDC gene was cloned into the vector and the resulting plasmid was transferred into *L. casei* as previously described[42]. Successful insertion of the TDC gene into the *L. casei* was confirmed by PCR and the production of tryptamine when cultured in an MRS broth containing 0.25% tryptophan. The engineered *L. casei* TDC+ and vector-only *L. casei* were grown on MRS agar plates containing erythromycin (50 μg/mL) and further grown in MRS broth. The *L. casei* TDC+ and vector-only For administration, *L. casei* were collected from the medium by centrifugation at 3000 rpm for 10 min at room temperature. *L. casei* inoculums were prepared in 300 μL of sterilized PBS and then administered to antibiotics-treated mice by oral gavage.

Primers used for determining TDC plasmid are (F1: CGGTCCTCGGGATATGATAAGA; R1: GACCCTCCGCTTACAAAGAC).

Primers used for Sanger sequencing are (F1: CGGTCCTCGGGATATGATAAGA; R1: GACCCTCCGCTTACAAAGAC; R2: AGGCAGCTGATCTCAACAATG).

### Study methods details

**Reagents and resources.** Reagents and resources details are provided in (Supplementary Data 4).

**Fecal suspension administration.** For fecal samples of monkeys, about 10 g of fecal samples were mixed with 5× sterilized 1× phosphate-buffered saline (PBS, m/v) and homogenized as fecal suspension. HFD-fed mice (8 weeks HFD treatment) were orally administered with the fecal suspension derived from normal and diabetic monkeys at 4 g/kg daily for 5 days according to the physiological range of tryptamine and phenethylamine we detected in monkeys. On day 5, following a 12-h fast, an OGTT was performed to examine the effects of the fecal suspension from monkeys on glucose tolerance in recipient mice.

**Fecal microbiota transplantation.** For fecal samples of humans, about 2 g of fresh fecal samples were mixed with 5× sterilized 1× phosphate-buffered saline (PBS, m/v) and homogenized as fecal microbiota suspension. Antibiotics-treated mice were orally administered with the 300 μL fecal microbiota suspension for 5 days. On day 3, following a 12-h fast, an OGTT was performed to examine the effects of the fecal microbiota from humans on glucose tolerance in recipient mice.

**Metabolomics study.** About 150 mg of feces were extracted with a 20× volume of 70% methanol (m/v) and then homogenized with steel beads. The samples were then centrifuged at 14, 000 rpm at 4 °C for 15 min. About 200 μL of the supernatant was transferred to new tubes for LC-MS analysis. A pooled quality control (QC) sample was prepared by mixing equal amounts of each sample. An Agilent UPLC system coupled to a triple quadrupole (QQQ) 6460 mass spectrometry was used for targeted metabolomics. A Waters BEH 2.1×100 mm C18 1.7 μm column with a pre-column was used. The mobile phase used in LC-MS-QQQ was A: water with 0.1% formic acid and B: acetonitrile with 0.1% formic acid. The gradients were set as 2% B (0–0.5 min), 2–30% B (0.5–4 min), 30–100% B (4–6 min), 100% B (6–8 min), 100–2% B (8–8.1 min) and maintained in 2% B (8.1–10 min). The standards list, MRM transition and retention time are provided in (Supplementary Data 5). LC-MS data were collected and analyzed using Agilent MassHunter Workstation Software.

**Batch culture of fecal samples.** About 50 mg of fecal samples were mixed with 20× volume of sterilized 1× PBS (m/v) and homogenized with steel beads. The fecal suspension (20 μL) was inoculated in 2 mL Tryptic Soy Broth (TSB) supplemented with 0.25% tryptophan and incubated overnight under anaerobic conditions at 37 °C. After incubation, 100 μL of the medium was then used and processed for quantification of tryptamine by LC-MS analysis following the serum protocol of LC-MS.

**Glucose and insulin tolerance test.** For the oral glucose tolerance test (OGTT), mice were fasted for 12 h (overnight) and administered with tryptamine, ERK inhibitors, TAAR1 antagonist or IAA at indicated dosages. After 30 min, mice were given glucose at a dosage of 2 g/kg. Blood samples were collected from the tail vein for glucose measurement using Accu-Chek glucose meters at 0, 15, 30, 60, 90, and 120 min after the glucose challenge. For the insulin tolerance test (ITT), mice were fasted for 4 h and administered with tryptamine, ERK inhibitors, or a TAAR1 antagonist at indicated dosages. After 30 min, insulin (1 U/kg) was injected intraperitoneally into the mice. Blood glucose levels were measured as per OGTT. For the measurement of serum insulin and TG, serum was collected 120 min after the mice were treated with tryptamine.

**Tissue distribution of tryptamine.** Tryptamine at a dosage of 5 mg/kg (dissolved in 0.5% CMC-Na) was orally administered to mice. After 15 min, mice were euthanized by isoflurane and sacrificed by cervical dislocation. Serum, liver, skeletal muscle, and WAT were collected and stored at −80 °C until analysis. About 80 mg of biological tissues were extracted with a 20× volume of 70% methanol (m/v) and then homogenized with steel beads. The samples were then centrifuged at 14, 000 rpm at 4 °C for 15 min. About 200 μL of the supernatant was transferred to new tubes for LC-MS analysis.

**Phospho-proteomics study.** Mice were fasted for 4 h and orally administered with tryptamine or vehicle. The WAT tissues were collected for the phosphoproteomics study. Mice tissue samples used for the TMT-labeled phosphoproteomics study were prepared as previously reported[43]. An Easy nLC system (Thermo Fisher Scientific) with an Acclaim PepMap RSLC column (50 μm × 15 cm) was used to separate the TMT-labelled peptides. The mobile phase used in LC-MS-

Orbitrap was A: water with 0.1% formic acid and B: 80% acetonitrile, 20% water with 0.1% formic acid. The elution gradient was set as 0–5 min in 0-6% buffer B, 5–45 min in 6–28% buffer B, 45–50 min in 28%–38% buffer B, 50–55 min 38–100% buffer B and maintained during 50–60 min in 100% buffer B. The obtained MS/MS spectra were processed by Proteome Discoverer (Thermo Fisher Scientific) and searched using MASCOT engine 2.6. All protein sequences were aligned to the *Mus musculus* database downloaded from UniProt (http://www.uniprot.org). The proteins with a fold change >1.2 or <0.8 and a p-value < 0.05 were considered as differentially expressed proteins. KEGG pathway annotation was performed using KOALA (KEGG Orthology And Links Annotation) to identify the significantly enriched pathways.

**Protein analysis.** Frozen tissues and harvested cells were lysed in RIPA buffer with a protease inhibitor cocktail. For western blotting, the cell lysates and tissue lysates were centrifuged at 15,000 rpm for 15 min at 4 °C. The supernatant was mixed with 5× loading buffer and heated at 98 °C on a dry bath for 10 min. The target proteins were then detected in the samples as per manufacturer instructions. The blots were incubated with HRP-linked anti-rabbit IgG or anti-mouse IgG and reacted with enhanced chemiluminescence. The quantification of protein bands from western blots was analyzed using Image J.

**Characterization of tryptamine-producing bacteria.** The two reference tryptophan decarboxylase sequences of *Ruminococcus gnavus* (strain ATCC 29149/VPI C7-9) and *Clostridium sporogenes* (strain ATCC 15579) were downloaded from ENA (A7B1V0 and J7SZ64)[16]. The identity between these two reference sequences was 26% based on BLASTP. To identify potential tryptophan decarboxylase sequences in the GUT2D dataset, BLASTP was used to align the two reference sequences against the nonredundant microbiome gene catalog constructed in the GUT2D study[20]. The alignments were filtered with E-value < 1e-5 and identity >30%. Repeated measures of correlation coefficients between fecal metabolites and abundances of the 5 co-abundance groups were calculated as previously described[44].

## Quantification and Statistical analysis

Data are expressed as average and SD or SEM values of at least triplicates. P-values were calculated using GraphPad Prism 8 and p-values less than 0.05 are considered statistically significant. Wilcoxon rank-sum test (one-tailed or two-tailed test) was employed to determine the differences in metabolomics data between subjects with and without T2D. Unpaired Student's t-tests or one-way ANOVA were employed in other settings as indicated. For clinical research studies. the statistical method and randomization method can be found in the study protocols mentioned above accordingly. For basic research studies, no statistical method was used to predetermine sample size and no data were excluded from the analyses. The experiments are not randomized, and investigators were not blinded to allocation during experiments and outcome assessment.

### Reporting summary

Further information on research design is available in the Nature Portfolio Reporting Summary linked to this article.

## Data availability

All data supporting the findings in this study are available within the Article and Supplementary Information. Source data are provided as a Source Data file with this paper. This study generated a *Lactobacillus casei* bacteria strain that can produce phenethylamine and tryptamine. The vector control and engineered *L.casei* bacteria strain can be obtained via Zhao-Xiang Bian (bzxiang@hkbu.edu.hk). Fecal metagenomic sequencing data of the IBS study[40] can be obtained from CNGB Nucleotide Sequence Archive (https://db.cngb.org/cnsa/) under

accession number CNP0000334. Fecal metagenomic sequencing data of GUT2D study[20] can be obtained via the European Nucleotide Archive under accession numbers PRJEB14155. The phospho-proteomics data are available via ProteomeXchange with the identifier PXD044161. Further information and other requests for resources and reagents can also be directed to and will be fulfilled by Zhao-Xiang Bian.

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

## Acknowledgements

This work was supported by the Health@InnoHK Initiative Fund of the Hong Kong Special Administrative Region Government (ITC RC/IHK/4/7 to ZX.B), the Key-Area Research and Development Program of Guangdong Province (2020B1111110003 to ZX.B.), the Open Research Project Programme of the State Key Laboratory of Quality Research in Chinese Medicine (University of Macau) (SKL-QRCM-OP21008 to CY.L.) and the Shenzhen Science and Technology Innovation Committee (JCYJ20190808164201654 to HT.X.). We also thank the Vincent and Lily Woo Foundation for their support. We offer thanks to all patients and healthy volunteers who donated specimens for this study.

## Author contributions

Conceptualization: Z.X.B., L.P.Z., W.J., and J.Y.N.L. Methodology: L.X.Z., H.T.X., Y.Y.L., and H.L.X.W. Investigation: L.X.Z., H.T.X., C.Y.L., M.X.G., Z.W.N., G.J.W., Y.S.D., C.Y., C.H.H., Y.J.Z., and J.Y.L. Visualization: L.X.Z., C.Y.L., and H.L.X.W. Funding acquisition: H.T.X. and Z.X.B. Project administration: L.X.Z., J.Y.L., Y.Y.L., and H.L.X.W. Supervision: L.Zhang, L.Zhao, C.H.Z., A.P.L., and L.T.L. Writing- original draft: L.X.Z., H.T.X., C.Y.L., and H.L.X.W. Writing- review & editing: Z.X.B., L.P.Z., W.J., J.Y.N.L., and Y.Y.L.

## Competing interests

The authors declare no competing interests.

## Additional information

[1]Centre for Chinese Herbal Medicine Drug Development, Hong Kong Baptist University, Hong Kong SAR, China. [2]School of Chinese Medicine, Hong Kong Baptist University, Hong Kong SAR, China. [3]School of Pharmaceutical Sciences, Health Science Center, Shenzhen University, Shenzhen, China. [4]State Key Laboratory of Microbial Metabolism and Ministry of Education Key Laboratory of Systems Biomedicine, School of Life Sciences and Biotechnology, Shanghai Jiao Tong University, Shanghai, China. [5]Department of Biochemistry and Microbiology and New Jersey Institute for Food, Nutrition, and Healthy. School of Environmental and Biological Sciences, Rutgers University, New Brunswick, NJ 08901, USA. [6]Department of Computer Science, Hong Kong Baptist University, Hong Kong SAR, China. [7]Academy of Integrative Medicine, Shanghai University of Traditional Chinese Medicine, Shanghai, China. [8]Phenome Research Centre, School of Chinese Medicine, Hong Kong Baptist University, Hong Kong SAR, China. [9]Shanghai Key Laboratory of Diabetes Mellitus and Center for Translational Medicine, Shanghai Jiao Tong University Affiliated Sixth People's Hospital, Shanghai, China. [10]These authors contributed equally: Lixiang Zhai, Haitao Xiao, Chengyuan Lin, Hoi Leong Xavier Wong, Yan Y. Lam. ✉e-mail: weijia1@hkbu.edu.hk; liping.zhao@rugters.edu; bianzxiang@gmail.com

