## [Peer Review File · Nature Communications]

REVIEWER COMMENTS

Reviewer #1 (Remarks to the Author):

COMMENTS:

Gut dysbiosis and consequent functional alterations of the gut microbiome are recently recognized as novel factors in disease causation. Comparison of systemic and fecal metabolomic profiles in healthy controls (exhibiting eubiosis) and diseased populations (exhibiting dysbiosis) has provided unique information on gut microbially derived metabolites responsible for respective phenotypes. Using a similar approach, in the current article, Zhai and group have described a role of gut microbially derived trace amines in IBS associated metabolic syndrome. Elevated levels of trace amines, specifically tryptamine (TM) and phenethylamine (PEA), correlated significantly with insulin resistance index in IBS and T2D cohorts, and with a specific gut microbe *R. gnavus*. Fecal microbial transfer from IBS/diabetic monkeys, mono-association of germ-free mice with this strain or ectopic expression of microbial decarboxylase enzyme via *L. casei* produced metabolic syndrome like features in recipient mice with high fecal/plasma levels of TM and PEA. These effects were found to be dependent on trace amine associated receptor 1 (TAAR1) activity and were abolished when the receptor was antagonized or knocked out. This is an interesting study with strengths of parallel use of clinical and pre-clinical models, and retrograde translation and mono-association models to establish causation.

Some Major concerns diminish my enthusiasm as follows,

- 1) It feels as though the authors are trying to do too much. By including IBS, metabolic syndrome, and diabetes in two animal models and a human patient cohort, the discussion and the analysis seems to jump around a lot without substantially proving some of their proposed hypothesis. Maybe focusing on IBS or metabolic disease would help focus the paper.
- 2) Serotonin levels vary in IBS subjects. Serotonin has been shown to modify insulin resistance and even act as a weak agonist for TAAR1. Thus, it will be important to assess this pathway in the models used in the study.
- 3) IBS or T2D FMT in TAAR1 knockout mice could signify the presence of this other pathway. Could gut inflammation as noted in IBS and T2D play a role in observed phenotype?
- 4) TAAR1 has been shown to increase GLP-1 and PYY secretion and modulate feeding behavior through neural effects. How GLP-1 levels differed in the studied models? TAAR1 expression and

activity in liver, adipose and muscle tissues is not well characterized. The results should be supplemented with TAAR1 expression data in these tissues.

5) Among characterized gut microbes, *R. gnavus* was found positively correlated with TyG ($r=0.203$, $p<0.001$, Figure.1D) in HC and IBS patients and enriched in IBS patients with higher TyG index. The linear model would be more convincing if they ran the association with HC and IBS patients separately. It seems as though *R. gnavus* has a positive association with all participants as there are no discernable visual differences between IBS and HC patients. They report different models (2 r values) for the tryptamine and phenethylamine models in the next sentence, which leads me to question why they did not run two models for the other association as well.

6) Administration of tryptamine and phenethylamine -The range found in fecal samples likely don't match intake or levels absorbed and are likely not within physiologic range when injected intraperitoneally. I would also be interested to know how it compares to the baseline levels in mice or the levels induced by *R. gnavus*?

7) They discuss the monkeys spontaneous development of metabolic syndrome but do not have any experimental models looking at the microbiome or tryptamine and phenethylamine levels in metabolic syndrome in this model.

8) Is *R. gnavus* present in the diabetic monkey metabolome? Or is a different bacteria driving the production of tryptamine and phenethylamine? I wish they had sequenced the monkey microbiome and shown the differentially abundant bacteria present.

Minor:

1. Line #77-79 Study reference is not included.
2. Line # 116-118 Unclear if systemic or fecal levels of TM and PEA were determined.
3. Figure 5E. Fecal levels of TM between normal and prediabetic monkeys does not appear significantly different.
4. Line# 285-287 Tyramine levels have been found to be significantly different in IBS cohort in the referenced study [[doi:https://doi.org/10.1016/j.chom.2022.11.006](https://doi.org/10.1016/j.chom.2022.11.006)].
5. Line# 289-290 There is no direct assessment of insulin secretion in the current study.
6. Phosphorylation of Thr308 of akt is not included.
7. Methods: (a) fecal microbial transplant procedure needs to be reconciled with respect to results section (was dietary treatment given to the mice?), (b) how the dose was determined is unclear (4g/kg?), (c) how were human to mouse fecal transplants performed? (d) what control groups were included in FMT experiments? and (e) reconcile if all insulin sensitive tissues or only WAT was used for phospho-proteomics study.
8. Which animal model is used on pg. 5?

9. They discuss seeing an increase in IAA, but also note in the previous paragraph no effects of IAA
10. I am confused by the ERK inhibition work. They discuss using U0126, but then mention multiple inhibitors. Also, there is no discussion of inhibition of different steps of the MAPK/ERK pathway
11. The monkey model is mentioned on line 233, but experiments are discussed above ~ line 190
12. The effect of diabetic monkey microbiome on mouse glucose levels should probably be discussed earlier. A lot of the previous work is based on the understanding that the microbiome and microbial metabolites are driving glucose intolerance.
13. The fiber models should control for sex and age as well as BMI.

Reviewer #2 (Remarks to the Author):

The manuscript "Gut microbiota-derived tryptamine and phenethylamine impair insulin sensitivity in metabolic syndrome and irritable bowel syndrome" from Zhai L et al. investigates the role of microbially derived tryptamine and phenylethylamine in insulin resistance and metabolic syndrome. Authors systematically dissect the effect of tryptamine and phenylethylamine using multiple models including gnotobiotic and antibiotic treated mice, primate models and correlate their findings in human subjects. Overall, there is sufficient data to justify the conclusions, however there are significant concerns regarding the methods and choice of models for some of the experiments as outlined below which should be addressed to ensure rigor.

1. Authors use *R. gnavus* as the native strain harboring tryptamine decarboxylase and as it is not amenable to genetics, they engineer *L. casei* to express TDC. The engineered strain is an ideal way to specifically study the effect of bacterially derived metabolites and WT *L. casei* serves as a rigorous control strain which only differs in its ability to produce tryptamine. However, these strains are not used for majority of the experiments. Most of the effects described in the manuscript rely on systemic circulation of tryptamine, hence for the experiment using WT and engineered *L. casei*, authors should report plasma tryptamine levels, but only stool tryptamine levels are reported- this is also important to determine if in the subsequent experiments if the doses of pure compounds are appropriate and result in comparable levels of circulating metabolites as seen with WT and engineered *L. casei*.
2. As outlined above, authors do not provide a good explanation for administering pure compound rather than using *L. casei* which would have allowed better assessment of the effects of bacterially produced metabolites. It is unclear how the doses were determined and why the highest dose is used for some of the molecular experiments. It is difficult to extrapolate if similar results would be seen with bacterially produced tryptamine and phenylethylamine.
3. Authors do not provide detailed methods for antibiotic treatment model – how long were the antibiotics administered? how long were the bacteria administered after antibiotic treatment? Was

antibiotic treatment continued after gavaging with test bacteria. Also, there is no data on baseline microbial composition and levels of tryptamine and phenylethylamine after antibiotic assessment and the effect of test strains on microbial composition.

4. Authors have previously described activation of TAAR1 by tryptamine and phenylethylamine and in this manuscript they link TAAR1 with ERK activation; it would have been helpful to delineate downstream mechanism from TAAR1 activation which is a Gs GPCR with ERK activation. In figure 4 control using EPPTB alone is missing – this is important to ensure that the effect is specific to R. gnavus mediated mechanism. Also it would have been better to use WT and engineered L. casei for this experiment given that it provides a controlled system.

5. Mice colonized with healthy control with high tryptamine and phenylethylamine do not show an effect on glucose levels.

6. In figure 5, only correlation of tryptamine and phenylethylamine with blood glucose or HbA1C is reported; it would help to see if there were other metabolites that were different and did or did not correlate similarly with blood glucose and HbA1C.

7. As there are different statistical tests used, it will help to list the specific statistical test used in the legend and clarify the sexes of mice used for each experiment.

8. There is no demographic/metadata/diet information provided for the clinical cohorts and the correlational analysis do not account for any confounding factors.

REVIEWER COMMENTS

Reviewer #1 (Remarks to the Author):

COMMENTS:

Gut dysbiosis and consequent functional alterations of the gut microbiome are recently recognized as novel factors in disease causation. Comparison of systemic and fecal metabolomic profiles in healthy controls (exhibiting eubiosis) and diseased populations (exhibiting dysbiosis) has provided unique information on gut microbially derived metabolites responsible for respective phenotypes. Using a similar approach, in the current article, Zhai and group have described a role of gut microbially derived trace amines in IBS associated metabolic syndrome. Elevated levels of trace amines, specifically tryptamine (TM) and phenethylamine (PEA), correlated significantly with insulin resistance index in IBS and T2D cohorts, and with a specific gut microbe *R. gnavus*. Fecal microbial transfer from IBS/diabetic monkeys, mono-association of germ-free mice with this strain or ectopic expression of microbial decarboxylase enzyme via *L. casei* produced metabolic syndrome-like features in recipient mice with high fecal/plasma levels of TM and PEA. These effects were found to be dependent on trace amine associated receptor 1 (TAAR1) activity and were abolished when the receptor was antagonized or knocked out. This is an interesting study with strengths of parallel use of clinical and pre-clinical models, and retrograde translation and mono-association models to establish causation.

Answer: Thank you very much for your encouraging comments. We sincerely appreciate your valuable time in reviewing our manuscript. We have addressed your concerns and comments point-by-point as follows:

Some Major concerns diminish my enthusiasm as follows,

1) It feels as though the authors are trying to do too much. By including IBS, metabolic syndrome, and diabetes in two animal models and a human patient cohort, the discussion and the analysis seems to jump around a lot without substantially proving some of their proposed hypothesis. Maybe focusing on IBS or metabolic disease would help focus the paper.

Answer: Thank you for your comments. To further clarify our research target and strengthen the conclusion, we revised the structure of this study by adjusting the order of clinical data and basic research data. We first presented the clinical evidence of elevated levels of trace amines and related trace amine producers in both IBS and T2D and then demonstrated their inhibitory effects on insulin sensitivity.

Besides, we have also revised the discussion to help focus the paper.

Please find the revised content from Page 6, Line 118 to Page 14, Line 348.

2) Serotonin levels vary in IBS subjects. Serotonin has been shown to modify insulin resistance and even act as a weak agonist for TAAR1. Thus, it will be important to assess this pathway in the models used in the study.

Answer: We highly agree with your comment as serotonin plays an important role in glucose and lipid metabolism. Inhibition of serotonin biosynthesis has been shown to improve metabolic syndrome, insulin resistance and nonalcoholic fatty liver disease in mice (Nature Medicine, 2015 DOI: 10.1038/nm.3766) whereas serotonin activation improves beta cell survival by stimulating their proliferation (Science Translational Medicine, 2020, DOI: 10.1126/scitranslmed.aay0455.). Moreover, our previous study has shown tryptamine and phenethylamine stimulate the biosynthesis of peripheral serotonin to induce diarrhea-like behavior in mice (Cell Host & Microbe, 2023, DOI: 10.1016/j.chom.2022.11.006). To explore whether serotonin is indeed involved in the impaired insulin sensitivity induced by tryptamine and phenethylamine, we examined the impact of serotonin inhibition on insulin-desensitizing effect of tryptamine and phenethylamine. We compared the inhibitory effects of tryptamine and phenethylamine on insulin sensitivity in normal mice and mice treated with an inhibitor of TPH1, a key enzyme for serotonin biosynthesis).

In our previous study, we showed inhibition of TPH1 using chemical inhibitor LX-1031 significantly suppressed the stimulatory effects of tryptamine and phenethylamine on serotonin production (Cell Host & Microbe, 2023, DOI: 10.1016/j.chom.2022.11.006). We, therefore, used LX-1031, as an inhibitor of tryptophan decarboxylase 1 (TPH1), to block the stimulatory effects of tryptamine and phenethylamine on serotonin production.

In *in vivo* study, we showed inhibition of serotonin synthesis in mice by LX-1031 did not affect the inhibitory effects of tryptamine on glucose tolerance, suggesting that acute treatment of tryptamine regulate insulin sensitivity in a serotonin-independent manner.

In parallel with this *in vivo* finding, we showed tryptamine/phenethylamine directly impaired insulin signaling in a time-dependent manner in 3T3-L1 adipocytes, an *in vitro* cell line model that does not produce serotonin. Therefore, we concluded that the effects of tryptamine/phenethylamine on insulin sensitivity is not mediated by serotonin production.

However, as inhibition of peripheral serotonin biosynthesis reduces obesity and metabolic dysfunction (Nature medicine, <https://doi.org/10.1038/nm.3766>), and release of serotonin from EC cells of gut into the circulation increases the obesity and type 2 diabetes (Endocr Rev, 10.1210/er.2018-00283), serotonin plays vital role in regulating insulin sensitivity and drives the development of insulin resistance. Therefore, the stimulation of serotonin by tryptamine/phenethylamine may also contribute to the development of insulin resistance in the long-term.

Please find this data in Figure.S6H. and updated content in Page 17, line 350-361.

3) IBS or T2D FMT in TAAR1 knockout mice could signify the presence of this other pathway. Could gut inflammation as noted in IBS and T2D play a role in observed phenotype?

Answer: Thanks for your excellent suggestions. To examine whether tryptamine and phenethylamine would induce inflammation, we treated the RAW 264.7 cells, a mouse/human macrophage cell line, with tryptamine. We found that the RAW 264.7

cells did not show any sign of inflammation after the treatment of tryptamine. In contrast, tryptamine suppressed the pro-inflammatory responses of LPS-treated RAW 264.7 cells as measured by nitric oxide level (Figure below), which was consistent with the previous study showing the anti-inflammatory effect of two amines *in vitro*. (Int J Mol Sci. doi: 10.3390/ijms231911209.)

These data suggest that insulin resistance induced by bacteria-derived tryptamine and phenethylamine is unlikely associated with gut inflammation. Besides, gut inflammation is mainly caused by LPS derived from gut microbes. Interestingly, the producers of tryptamine and phenethylamine identified in this study including *Ruminococcus gnavus*, *Blautia hansenii* and *Clostridium bolteae* are all gram-positive bacteria that do not produce LPS. Moreover, the *Lactobacillus casei* that is used for the ectopic expression of microbial decarboxylase enzyme is also gram-positive. Therefore, the effects of tryptamine and phenethylamine-producing bacteria on insulin resistance are likely mediated by trace amines but not gut inflammation caused by LPS.

To further validate our findings, we performed the FMT experiments with IBS samples with high insulin resistance index in TAAR1 KO mice as shown in our revised manuscript. We found that TAAR1 ablation restored the glucose tolerance of mice transplanted with the samples with high insulin resistance index to the level of mice with the samples with low insulin resistance index, suggesting that the effects of tryptamine and phenethylamine-producing bacteria on insulin resistance are predominately mediated by TAAR1.

Please find these results in Figure. 7E (OGTT) and Figure.S7B (tryptamine and phenethylamine levels in fecal samples donors) and updated content in Page 14, Line 291-294

As the fecal samples of T2D patients have been used up in our previous experiments, we unfortunately could not perform the FMT experiment with T2D samples. We may consider to use fecal samples of diabetic monkeys as a substitute approach.

4) TAAR1 has been shown to increase GLP-1 and PYY secretion and modulate feeding behavior through neural effects. How GLP-1 levels differed in the studied models? TAAR1 expression and activity in liver, adipose and muscle tissues is not well characterized. The results should be supplemented with TAAR1 expression data in these tissues.

Answer: Thanks for your kind suggestion. We have determined the serum GLP-1 and PYY levels in tryptamine and phenethylamine-treated mice and we found that treatments with tryptamine and phenethylamine had negligible effects on GLP-1 or PYY levels in mice.

Please find this result in Figure.S4H-I and the updated content in page 12, line 227-229.

We also determined the expression level of *Taar1* in liver, adipose and muscle tissues of wildtype and *Taar1*^{-/-} mice using RT-PCR. In line with our findings, *Taar1* is expressed in insulin-sensitive tissues including liver, adipose and muscle of wildtype mice but not in *Taar1*^{-/-} mice.

Please find this result in Figure.6G.

More importantly, we found that TAAR1 ablation significantly abolished the insulin-desensitizing effect of tryptamine/PEA in these insulin-sensitive tissues

Please find this result in Figure.6H-K.

5) Among characterized gut microbes, *R. gnavus* was found positively correlated with TyG ($r=0.203$, $p<0.001$, Figure.1D) in HC and IBS patients and enriched in IBS patients with higher TyG index. The linear model would be more convincing if they ran the association with HC and IBS patients separately. It seems as though *R. gnavus* has a positive association with all participants as there are no discernable visual differences between IBS and HC patients. They report different models (2 r values) for the tryptamine and phenylamine models in the next sentence, which leads me to question why they did not run two models for the other association as well.

Answer: We have separated the HC and IBS cohorts and performed the correlation analysis between tryptamine and phenethylamine producers including *Ruminococcus gnavus* with TyG index.

Please find this result in Figure.1 D-E.

Our results showed both tryptamine and phenethylamine are positively correlated with TyG index in HC and IBS patients. Due to the difference in sample size of HC and IBS, the result can be further validated in a large cohort of HC subjects in the future.

6) Administration of tryptamine and phenethylamine -The range found in fecal samples likely don't match intake or levels absorbed and are likely not within physiologic range when injected intraperitoneally. I would also be interested to know how it compares to the baseline levels in mice or the levels induced by *R. gnavus*?

Answer: This is an important suggestion. For *in vivo* studies, based on the fact that the tryptamine and phenethylamine levels found in human fecal samples are about ~ 0.5 - $1\mu\text{g/g}$ (Figure.2A-B), the actual content of tryptamine and phenethylamine in the human gut are about 3-5 mg considering human has 3-5 kg luminal content in the gut. The dosage used for tryptamine and phenethylamine studies in mice would be 1-5

mg/kg. Therefore, we treated mice with tryptamine and phenethylamine starting from 0.4 mg/kg, 2mg/kg to 10 mg/kg in a dose-dependent manner. We also checked the serum levels of tryptamine after the injection and confirmed that concentration of tryptamine and phenethylamine was within the physiological range (Figure.S5A-B).

For in vitro studies, we have shown that tryptamine and phenethylamine inhibited insulin signaling in 3T3-L1 cells with the starting concentration of 10 μ M after 30 mins of treatment in a dose-dependent manner. Because insulin action can only be maintained for 60 mins in 3T3-L1 cells, we, therefore, used higher concentrations of tryptamine and phenethylamine (25 and 50 μ M) to trigger more significant action in insulin signaling with a relatively short period of incubation time (Figure.4I-L).

For the colonization study, we measured the tryptamine and phenethylamine levels in fecal samples of germ-free mice after colonization of *R. gnavus*, showing that colonization of *R. gnavus* resulted in elevated intestinal tryptamine and phenethylamine levels (10-20 μ g/g) within the physiological range we determined in human feces (1-5 μ g/g). Whereas in germ-free mice without the colonization of *R. gnavus*, the tryptamine and phenethylamine levels in feces were much lower than the basal levels in conventional mice (0.1-0.2 μ g/g) (Figure.1H-I and Figure.2A-B).

Collectively, our result showed administration of tryptamine and phenethylamine at indicated dosage or colonization of *R. gnavus* can reach the physiological range we determined in patients with T2D.

7) They discuss the monkeys spontaneous development of metabolic syndrome but do not have any experimental models looking at the microbiome or tryptamine and phenethylamine levels in metabolic syndrome in this model.

Answer: We have determined the gut microbiome and tryptamine and phenethylamine levels in monkeys.

Please find the tryptamine and phenethylamine levels in monkeys in Figure.2 E-F.

Please also find the microbiome data of experimental monkeys in the following figure.

D Correlation-s-pedes level

E

However, as we are preparing another study that includes the microbiome data of monkeys, we therefore do not publish this data in this study. The present study focuses on the role of gut microbiota-derived tryptamine and phenethylamine in the development of insulin resistance. Therefore the microbiome data of monkeys may not make significant contributions to the novelty of this study.

8) Is *R. gnavus* present in the diabetic monkey metabolome? Or is a different bacteria driving the production of tryptamine and phenethylamine? I wish they had sequenced the monkey microbiome and shown the differentially abundant bacteria present.

Answer: We have determined the abundances of potential tryptamine and phenethylamine producers and analyzed the correlation between these bacteria and glucose intolerance index. According to our results, *R. gnavus* was detected in the diabetic monkey metagenome, but we are not sure it is the same strain as seen in human studies as the current shot-gun metagenomics approach is not able to identify bacteria in strain level. Therefore, the bacterial strain that contributes to the production of tryptamine and phenethylamine in monkeys remains elucidated and would require further investigations such as strain isolation in the future.

Minor:

1. Line #77-79 Study reference is not included.

Answer: We have added the study reference accordingly.

Please find the reference in the updated manuscript line 81.

2. Line # 116-118 Unclear if systemic or fecal levels of TM and PEA were determined.

Answer: Fecal levels of tryptamine and phenethylamine were used for the correlation analysis. We have added this information to the revised manuscript.

Please find the updated information in line 118.

3. Figure 5E. Fecal levels of TM between normal and prediabetic monkeys does not appear significantly different.

Answer: We are sorry for our mistake of mislabeling. The significant elevation in TM levels was only observed in diabetic monkeys when compared with normal and prediabetic monkeys.

There is no significant difference in the fecal level of TM between normal and prediabetic monkeys.

Please find the updated Figure.5E in the updated manuscript.

4. Line# 285-287 Tyramine levels have been found to be significantly different in IBS cohort in the referenced study [doi:<https://doi.org/10.1016/j.chom.2022.11.006>].

Answer: In our previous study [<https://doi.org/10.1016/j.chom.2022.11.006>], we showed tyramine is also increased in IBS-D patients ($p < 0.05$) but not as significant as

tryptamine and phenethylamine ($p < 0.001$). We have updated this claim in the revised manuscript by only showing tyramine is not altered in diabetic monkeys and T2D subjects.

Please find the revised sentence in line 152-153 and line 171-172. Both fecal levels of tyrosine and tyramine results can be found in Figure.S2C-D and Figure.S2M-N.

5. Line# 289-290 There is no direct assessment of insulin secretion in the current study.

Answer: We measured the insulin levels in tryptamine-treated mice and monkeys and showed insulin levels were increased by the treatment of tryptamine and phenethylamine.

Please find these results in Figure.S4D and Figure.S4K. and updated content in Page 16, line 323-329.

6. Phosphorylation of Thr308 of akt is not included.

Answer: We have included the data showing the expression level of phosphorylation of Akt (Thr308) was also downregulated by tryptamine and phenethylamine in *in vivo* studies. Please find these results as follow.

7. Methods: (a) fecal microbial transplant procedure needs to be reconciled with respect to results section (was dietary treatment given to the mice?), (b) how the dose was determined is unclear (4g/kg?), (c) how were human to mouse fecal transplants performed? (d) what control groups were included in FMT experiments? and (e) reconcile if all insulin sensitive tissues or only WAT was used for phospho-proteomics study.

Answer: (a) We have revised the fecal microbial transplant procedure in the updated manuscript.

Please find the updated methods in page 22, line 464-476.

(b) The dose is determined based on the fecal levels of tryptamine and phenethylamine levels we determined in monkeys. As shown in Figure 5 E-F, fecal levels of tryptamine and phenethylamine range from 5 $\mu\text{g/g}$ to 15 $\mu\text{g/g}$. Therefore, there will be 5mg-15mg of tryptamine and phenethylamine in monkey feces according

to the luminal content in monkeys. Based on this concentration we detected, we gave mice monkey fecal suspension in a dosage of 4g/kg in order to reach a physiological range of tryptamine and phenethylamine in the mouse gut. We also determined the fecal level of tryptamine and phenethylamine and showed the concentration of tryptamine and phenethylamine (10-20ug/g) in fecal samples of recipient mice are within the physiological range.

(c) We have added the method for human-to-mouse fecal transplant experiments.

Please find the updated methods in page 23, line 472-476.

(d) In our present study we used WAT for phospho-proteomics analysis based on the results that phosphorylation of AKT in WAT is significantly inhibited by tryptamine and phenethylamine (table.s3). Furthermore, we showed the expression level of phosphorylation of ERK is significantly upregulated in all insulin-sensitive tissues by tryptamine treatment (Figure.4E).

8. Which animal model is used on pg. 5?

Answer: The mechanistic study involves normal wild type mice and Taar1 *KO* mice.

9. They discuss seeing an increase in IAA, but also note in the previous paragraph no effects of IAA

Answer: Our results showed tryptamine/ phenethylamine can be rapidly metabolized by the host into IAA (indole acetic acid) and phenylacetic acid. The increase of IAA is attributed to the increased levels of tryptamine.

To better understand the effects of tryptamine/phenethylamine on insulin signaling, we showed treatment with precursors and metabolites of tryptamine and phenethylamine including tryptophan, phenylalanine, indole-3-acetic acid and phenylacetic acid in similar doses did not alter the insulin-induced AKT phosphorylation in 3T3-L1 cells (Figure.S4F-G), indicating microbiota transformation of dietary tryptophan and phenylalanine plays an important role in the development of insulin resistance.

10. I am confused by the ERK inhibition work. They discuss using U0126, but then mention multiple inhibitors. Also, there is no discussion of inhibition of different steps of the MAPK/ERK pathway

Answer: In addition to ERK inhibitor U0126, we also used another ERK inhibitor PD98059 to address the role of the MAPK/ERK pathway in tryptamine and phenethylamine-induced insulin resistance (Figure.S5C and Figure.S5E)

Please find the updated content in Page 12, Line 246-257.

11. The monkey model is mentioned on line 233, but experiments are discussed above ~ line 190

Answer: We have adjusted the order of monkey data and experiments. We firstly introduce the positive association between tryptamine/phenethylamine and glucose

intolerance in type 2 diabetes including patients with type 2 diabetes and diabetic monkey models (Figure.2 and Figure.S2, line 145-179). Following that, we presented the causative data showing that tryptamine and phenethylamine weaken insulin signaling via TAAR1-ERK activation (Figure.4 and Figure.S4, line 201-229).

12. The effect of diabetic monkey microbiome on mouse glucose levels should probably be discussed earlier. A lot of the previous work is based on the understanding that the microbiome and microbial metabolites are driving glucose intolerance.

Answer: We have revised the discussion section to introduce the role of the microbiome and microbial metabolites of monkeys in the development of glucose intolerance.

Please find the updated content in Page 15, line 304-317.

13. The fiber models should control for sex and age as well as BMI.

Answer: The fiber study is designed and randomized by adjusting sex, age and BMI factors (Science, DOI: 10.1126/science.aao5774). We have also analyzed the data by adjusting sex, age and BMI factors following our previous study and updated the data in the revised manuscript.

Please find the updated content in Figure.3 and Figure.S3.

Reviewer #2 (Remarks to the Author):

The manuscript “Gut microbiota-derived tryptamine and phenethylamine impair insulin sensitivity in metabolic syndrome and irritable bowel syndrome” from Zhai L et al. investigates the role of microbially derived tryptamine and phenylethylamine in insulin resistance and metabolic syndrome. Authors systematically dissect the effect of tryptamine and phenylethylamine using multiple models including gnotobiotic and antibiotic treated mice, primate models and correlate their findings in human subjects. Overall, there is sufficient data to justify the conclusions, however there are significant concerns regarding the methods and choice of models for some of the experiments as outlined below which should be addressed to ensure rigor.

Answer: Thank you very much for the encouraging comments. We have carefully considered your concerns and suggestions regarding our study design including methods and animal models. Based on your comments, we have revised the manuscript for your consideration.

1. Authors use *R. gnavus* as the native strain harboring tryptamine decarboxylase and as it is not amenable to genetics, they engineer *L. casei* to express TDC. The engineered strain is an ideal way to specifically study the effect of bacterially derived metabolites and WT *L. casei* serves as a rigorous control strain which only differs in its ability to produce tryptamine. However, these strains are not used for majority of the experiments. Most of the effects described in the manuscript rely on systemic circulation of tryptamine, hence for the experiment using WT and engineered *L. casei*, authors should report plasma tryptamine levels, but only stool tryptamine levels are reported- this is also important to determine if in the subsequent experiments if the doses of pure compounds are appropriate and result in comparable levels of circulating metabolites as seen with WT and engineered *L. casei*.

Answer: Thank you for your critical comments and we have updated the data of serum and fecal levels of tryptamine and phenethylamine in mice colonized with either WT and engineered TDC+ *L. casei* (Figure.1L-M). We found that their serum levels in mice colonized with engineered TDC+ *L. casei* were comparable to the levels observed in germ-free mice colonized with *Ruminococcus gnavus*, a producer of tryptamine and phenethylamine, and to the levels of mice with systemic treatment of tryptamine.

We also considered the dosage of administration of tryptamine and phenethylamine based on the concentration we determined in the patients with IBS and T2D.

For *in vivo* studies, based on the fact that the tryptamine and phenethylamine found in human fecal samples are about ~0.5-1 μ g/g (Figure.2A-B), the actual content of tryptamine and phenethylamine in the human gut are about 3-5 mg considering human has 3-5 kg luminal content in the gut. The dosage used for tryptamine and phenethylamine studies in mice would be 1-5 mg/kg. Therefore, we treated mice with tryptamine and phenethylamine starting from 0.4 mg/kg, 2mg/kg to 10 mg/kg in a dose-dependent manner. We also checked the serum levels of tryptamine after the injection and confirmed that the concentration of tryptamine and phenethylamine is within the physiological range (Figure.S5A-B).

For *in vitro* studies, we have shown that tryptamine and phenethylamine inhibited insulin signaling in 3T3-L1 cells with the starting concentration of 10 μ M after 30 mins of treatment in a dose-dependent manner. Because insulin action can only be maintained for 60 mins in 3T3-L1 cells, we, therefore, use higher concentrations of tryptamine and phenethylamine (25 and 50 μ M) in order to trigger significant action in insulin signaling with a relatively short period of incubation time (Figure.4I-L).

For the colonization study, we measured the tryptamine and phenethylamine levels in fecal samples of germ-free mice after colonization of *R. gnavus*, showing that colonization of *R. gnavus* resulted in elevated intestinal tryptamine and phenethylamine levels (10-20 μ g/g) within the physiological range we determined in human feces (1-5 μ g/g). Whereas in germ-free mice without the colonization of *R. gnavus*, the tryptamine and phenethylamine levels in feces were much lower than the basal levels in conventional mice (0.1-0.2 μ g/g) (Figure.1H-I and Figure.2A-B).

Collectively, our result showed administration of tryptamine and phenethylamine at indicated dosage or colonization of *R. gnavus* can reach the physiological range we determined in patients with T2D.

2. As outlined above, authors do not provide a good explanation for administering pure compound rather than using *L. casei* which would have allowed better assessment of the effects of bacterially produced metabolites. It is unclear how the doses were determined and why the highest dose is used for some of the molecular experiments. It is difficult to extrapolate if similar results would be seen with bacterially produced tryptamine and phenylethylamine.

Answer: The studies with pure compounds enabled us to assess the direct effects of tryptamine/PEA on insulin sensitivity *in vitro*, essential for the mechanistic studies of the tryptamine/PEA actions. In the subsequent *in vivo* studies, the experiments with either colonization of tryptamine and phenethylamine-producers or engineered bacteria could provide evidence as proof of concept that bacterially produced tryptamine/PEA can be a pathological factor for T2D. Although both types of experiments are equivalently important for dissecting the pathological role of tryptamine/PEA in T2D, the experiments with pure compounds would be more suitable for the mechanistic studies and therefore used for the majority of the experiments. Notably, the doses of pure compounds we used in our animal studies resulted in comparable levels of circulating metabolites and similar phenotypic changes in insulin sensitivity as seen with WT and engineered *L. casei*.

Please also find our answer in comment 1 to address the concentration and dosage of tryptamine and phenethylamine we used in *in vivo* and *in vitro* studies.

3. Authors do not provide detailed methods for antibiotic treatment model – how long were the antibiotics administered? how long were the bacteria administered after antibiotic treatment? Was antibiotic treatment continued after gavaging with test bacteria. Also, there is no data on baseline microbial composition and levels of tryptamine and phenylethylamine after antibiotic assessment and the effect of test strains on microbial composition.

Answer: Thank you for your critical comments. We have updated the methods for the antibiotic treatment model and provided more detailed experimental procedures.

Please find the updated content in Page 20 line 408-411 and Page 23 line 476-481.

We have addressed your comments as follows:

how long were the antibiotics administered?

Antibiotics-treated mice were generated using antibiotics cocktails containing 50mg/kg vancomycin, 100mg/kg neomycin, 100mg/kg metronidazole, 100mg/kg ampicillin, 50mg/kg streptomycin via oral gavage for 9 days (one time per day).

how long were the bacteria administered after antibiotic treatment?

Antibiotics-treated mice were orally administered with the 300 μ L fecal microbiota suspension for 5 days.

Was antibiotic treatment continued after gavaging with test bacteria?

Antibiotic cocktails administration was stopped 18 hours before the fecal microbiota transplantation.

There is no data on (1) baseline microbial composition and (2) levels of tryptamine and phenylethylamine after antibiotic assessment and (3) the effect of test strains on microbial composition.

Answer:

We have added the data of levels of tryptamine and phenylethylamine after antibiotic administration to show tryptamine and phenylethylamine levels (Figure.S11) is significantly reduced by antibiotics treatment and significantly increased in mice colonized with *R. gnavus* and fecal samples from IBS patients with high tryptamine levels.

Please find this updated result in the revised manuscript.

4. Authors have previously described activation of TAAR1 by tryptamine and phenylethylamine and in this manuscript they link TAAR1 with ERK activation; it would have been helpful to delineate downstream mechanism from TAAR1 activation which is a Gs GPCR with ERK activation. In figure 4 control using EPPTB alone is missing – this is important to ensure that the effect is specific to *R. gnavus* mediated mechanism. Also it would have been better to use WT and engineered *L. casei* for this experiment given that it provides a controlled system.

Answer:

(a) it would have been helpful to delineate downstream mechanism from TAAR1 activation which is a Gs GPCR with ERK activation

Following the reviewer's suggestions, we have added the references of the downstream mechanism of TAAR1 (GPCR)-ERK activation in the discussion (doi: 10.1074/jbc.RA118.005464.). TAAR1 triggers cAMP-mediated calcium influx and release, both of which are required for activation of a MAPK cascade utilizing calmodulin-dependent protein kinase II (CaMKII), Raf, and MAPK/ERK kinase 1/2 (MEK1/2).

(b) In figure 4 control using EPPTB alone is missing.

We have shown that EPPTB does not significantly affect glucose homeostasis in normal mice in the updated manuscript (Figure.6 and Figure.S6). Therefore, we did not add EPPTB control group in Figure.7 but instead use EPPTB+ R. gnnavus to address the effect is specific to the R. gnnavus mediated mechanism (Figure.7A-B).

Moreover, we have shown Taar1 KO mice are protected from glucose intolerance induced by fecal microbiota transplantation from IBS patients with high TyG index, suggesting Taar1 is an important target for the treatment of insulin resistance induced by gut dysbiosis.

5. Mice colonized with healthy control with high tryptamine and phenylethylamine do not show an effect on glucose levels.

Answer: The reason is that levels of tryptamine and phenylethylamine in healthy controls are not as high as in IBS patients. Please find the fecal levels of tryptamine and phenylethylamine we determined in the donors for this study (Figure.S7A).

6. In figure 5, only correlation of tryptamine and phenylethylamine with blood glucose or HbA1C is reported; it would help to see if there were other metabolites that were different and did or did not correlate similarly with blood glucose and HbA1C.

Answer: In our updated manuscript, we have shown that tryptophan/phenylalanine is not altered in patients with T2D and diabetic monkeys (Figure.S2A-B, Figure.S2K-L).

7. As there are different statistical tests used, it will help to list the specific statistical test used in the legend and clarify the sexes of mice used for each experiment.

Answer: We have specified the statistical tests used in the legend and provided the sexes of mice used for each experiment.

8. There is no demographic/metadata/diet information provided for the clinical cohorts and the correlational analysis do not account for any confounding factors.

Answer: The clinical studies are designed and randomized by adjusting sex, age and BMI factors. We have also analyzed the data by adjusting sex, age and BMI factors following our previous study and updated the data in the revised manuscript.

REVIEWERS' COMMENTS

Reviewer #1 (Remarks to the Author):

They addressed all my comments

Reviewer #2 (Remarks to the Author):

Thank you for extensive additional experiments.